# On the Role of Randomization
# in Adversarially Robust Classification

**Lucas Gnecco Heredia**[1]* **Muni Sreenivas Pydi**[1] **Laurent Meunier**[2]
**Benjamin Negrevergne**[1] **Yann Chevaleyre**[1]
[1] CNRS, LAMSADE, Université Paris Dauphine - PSL    [2] Payflows
{lucas.gnecco-heredia,muni.pydi}@dauphine.psl.eu
{benjamin.negrevergne,yann.chevaleyre}@lamsade.dauphine.fr
laurent@payflows.io

## Abstract

Deep neural networks are known to be vulnerable to small adversarial perturbations in test data. To defend against adversarial attacks, probabilistic classifiers have been proposed as an alternative to deterministic ones. However, literature has conflicting findings on the effectiveness of probabilistic classifiers in comparison to deterministic ones. In this paper, we clarify the role of randomization in building adversarially robust classifiers. Given a base hypothesis set of deterministic classifiers, we show the conditions under which a randomized ensemble outperforms the hypothesis set in adversarial risk, extending previous results. Additionally, we show that for any probabilistic binary classifier (including randomized ensembles), there exists a deterministic classifier that outperforms it. Finally, we give an explicit description of the deterministic hypothesis set that contains such a deterministic classifier for many types of commonly used probabilistic classifiers, *i.e.* randomized ensembles and parametric/input noise injection.

## 1  Introduction

Modern machine learning algorithms such as neural networks are highly sensitive to small, imperceptible adversarial perturbations of inputs [3, 43]. In the last decade, there has been a back-and-forth in research progress between developing increasingly potent attacks [5, 10, 19, 25] and the creation of practical defense mechanisms through empirical design [8, 30, 32]. In particular, probabilistic classifiers (also called stochastic classifiers or randomized classifiers) have been widely used to build strong defenses which can be roughly categorized into two groups: noise injection techniques [14, 28, 34, 37, 47, 48] and randomized ensembles [31, 38]. The first group includes methods that add noise at inference, usually to the input [22, 48], an intermediate layer activation [22, 47, 51] or the weights of a parametric model [22] like a neural network. Most of this work is experimental, with some theoretical papers that try to justify the use of such methods [37]. The second group, inspired by game theory, create a mixed strategy over base classifiers as a finite mixture [31, 38]. The argument behind this kind of models is that it becomes harder for an attacker to fool multiple models at the same time [12, 36].

Intuitively, the greater flexibility of probabilistic classifiers implies that they should outperform their deterministic counterparts in adversarially robust classification. For instance, one of the earliest works using randomization to improve robustness to adversarial attacks [48] claims that *"randomization at inference time makes the network much more robust to adversarial images"*. Dhillon et al. [14] propose a pruning method that *"is stochastic and has more freedom to deceive the adversary"*.

---

*Corresponding author

37th Conference on Neural Information Processing Systems (NeurIPS 2023).

Similarly, Panousis et al. [34] claim that for their method, *"the stochastic nature of the proposed activation significantly contributes to the robustness of the model"*.

Lending support to the intuition that probabilistic classifiers are better than their deterministic counterparts, Pinot et al. [38] prove the non-existence of a Nash equilibrium in the zero-sum game between an adversary and a classifier when both use deterministic strategies. Moreover, they show that a randomized ensemble outperforms any deterministic classifier under a regularized adversary. Meunier et al. [31] shows that a mixed Nash equilibrium does exist in the adversarial game when the classifier is allowed to use randomized strategies, making a strong case for probabilistic classifiers.

In contrast to the above results advocating for probabilistic classifiers, Pydi and Jog [40] use an equivalence between adversarial risk minimization and a certain optimal transport problem between data generating distributions to show that in the case of binary classification there exists an approximate pure Nash equilibrium in the adversarial game, showing that there always exists a sequence of deterministic classifiers that are optimal for adversarial risk minimization. Trillos et al. [45] extend their results to multiclass classification using a multimarginal optimal transport formulation. More recently, Awasthi et al. [1, 2] prove the existence of a deterministic classifier that attains the optimal value for adversarial risk under mild assumptions. Trillos et al. [44] extends it to the multiclass setting.

How does one reconcile these apparently conflicting theoretical findings as a machine learning practitioner? First, it is important to note that the above theoretical results are valid when considering large hypothesis sets, like that of all possible measurable classifiers. In practice, one works with more limited hypotheses sets such as the space of linear classifiers or the space of classifiers that can be learned by an $L$-layered deep neural network, for which the above theoretical results do not directly apply. Second, the task of training a robust probabilistic classifier is non-trivial [12, 39]. Even the task of efficiently attacking a probabilistic classifier is not straightforward [6, 12, 13, 36]. Hence, it is important to understand the precise role of randomization in adversarial robustness and the conditions under which probabilistic classifiers offer a discernible advantage over deterministic classifiers. In this work, we address this issue by making the following contributions.

- **From deterministic to probabilistic classifiers:** Given a base hypothesis set (BHS) $\mathcal{H}_b$ of deterministic classifiers, we prove the necessary and sufficient conditions for a probabilistic classifier built from $\mathcal{H}_b$ to strictly outperform the best classifier in $\mathcal{H}_b$ in adversarial risk. A quantity that plays a crucial role in our analysis is the *expected matching penny gap* wherein there exist classifiers that are vulnerable to adversarial perturbation at a point, but not all at the same time. These results are contained in Section 3.

- **From probabilistic to deterministic classifiers:** In the binary classification setting, given a probabilistic classifier **h**, we show that there always exists a deterministic classifier $h$ belonging to a "threshold" hypothesis set (THS) that is built from **h** in a straightforward way. We then apply our results to common families of probabilistic classifiers, leading to two important conclusions: 1) *For every randomized ensemble classifier [13, 38], there exists a deterministic weighted ensemble [17, 52] with better adversarial risk.* 2) *For every input noise injection classifier [22, 37, 39], there exists a deterministic classifier that is a slight generalization of randomized smoothing classifier [8, 26] with better adversarial risk.* These results are contained in Section 4.

- **Families of classifiers for which randomization does not help:** Given a BHS $\mathcal{H}_b$, we show the conditions under which randomizing over classifiers in $\mathcal{H}_b$ does not help to improve adversarial risk. Specifically, if the decision regions of $\mathcal{H}_b$ are closed under union and intersection, our results imply that randomized ensembles built using $\mathcal{H}_b$ offer no advantage over the deterministic classifiers in $\mathcal{H}_b$. These results are contained in Section 5.

**Notation:** We use $\mathbb{1}\{C\}$ to denote an indicator function which takes a value 1 if the proposition $C$ is true, and 0 otherwise. For an input space $\mathcal{X} \subseteq \mathbb{R}^d$, we use $\sigma(\mathcal{X})$ to denote some $\sigma$-algebra over $\mathcal{X}$ that makes $(\mathcal{X}, \sigma(\mathcal{X}))$ a measurable space. If not specified, $\sigma(\mathcal{X})$ will be the Borel $\sigma$-algebra, which is generated by the open sets of $\mathcal{X}$. We use $\mathcal{P}(\mathcal{X})$ to denote the set of probability measures over the measure space $(\mathcal{X}, \sigma(\mathcal{X}))$. For a positive integer $K$, we use $[K]$ to denote the range $\{1, \ldots, K\}$. For a vector $u \in R^K$, we denote $u_i$ the $i$th component of $u$.

## 2 Preliminaries

### 2.1 Adversarially Robust Classification

We consider a classification setting with input space $\mathcal{X} \subseteq \mathbb{R}^d$ and a finite label space $\mathcal{Y}$. The space $\mathcal{X}$ is equipped with some norm $\|\cdot\|$, which is commonly set to be the $\ell_2$ or $\ell_\infty$ norms. Let $\rho \in \mathcal{P}(\mathcal{X} \times \mathcal{Y})$ denote the true data distribution, which can be factored as $\rho(x, y) = \nu(y)\rho_y(x)$, where $\nu \in \mathcal{P}(\mathcal{Y})$ denotes the marginal probability distribution of $\rho$ over $\mathcal{Y}$ and $\rho_y \in \mathcal{P}(\mathcal{X})$ denotes the conditional probability distribution of the data over $\mathcal{X}$ given class $y$.

A *deterministic classifier* is a function $h : \mathcal{X} \to \mathcal{Y}$ that maps each $x \in \mathcal{X}$ to a fixed label in $\mathcal{Y}$. The 0-1 loss of $h$ on a point $(x, y) \in \mathcal{X} \times \mathcal{Y}$ is given by, $\ell^{0\text{-}1}((x,y), h) = \mathbb{1}\{h(x) \neq y\}$.

A *probabilistic classifier* is a function $\mathbf{h} : \mathcal{X} \to \mathcal{P}(\mathcal{Y})$ that maps each $x \in \mathcal{X}$ to a probability distribution over $\mathcal{Y}$. To label $x \in \mathcal{X}$ with $\mathbf{h}$, one samples a random label from the distribution $\mathbf{h}(x) \in \mathcal{P}(\mathcal{Y})$. In practice, if $\mathcal{Y}$ consists of $K$ classes, $\mathcal{P}(\mathcal{Y})$ is identifiable with the $K$-simplex $\Delta^K$ that consist of vectors $u \in \mathbb{R}^K$ such that $\sum_{i=1}^K u_i = 1$. Therefore, one can think of $\mathbf{h}(x)$ as a probability vector for every $x$.

The 0-1 loss of $\mathbf{h}$ on $(x, y) \in \mathcal{X} \times \mathcal{Y}$ is given by, $\ell^{0\text{-}1}((x,y), \mathbf{h}) = \mathbb{E}_{\hat{y} \sim \mathbf{h}(x)}[\mathbb{1}\{\hat{y} \neq y\}] = 1 - \mathbf{h}(x)_y$. Note that $\ell^{0\text{-}1}((x,y), \mathbf{h}) \in [0, 1]$ is a generalization of the classical 0-1 loss for deterministic classifiers, which can only take values in $\{0, 1\}$.

Given $x \in \mathcal{X}$, we consider a data perturbing adversary of budget $\epsilon \geq 0$ that can transport $x$ to $x' \in B_\epsilon(x) = \{x' \in \mathcal{X} \mid d(x, x') \leq \epsilon\}$, where $B_\epsilon(x)$ is the closed ball of radius $\epsilon$ around $x$. The adversarial risk of a probabilistic classifier $\mathbf{h}$ is defined as follows.

$$\mathcal{R}_\epsilon(\mathbf{h}) = \mathbb{E}_{y \sim \nu} \left[ \mathcal{R}_\epsilon^y(\mathbf{h}) \right] = \mathbb{E}_{y \sim \nu} \mathbb{E}_{x \sim p_y} \left[ \sup_{x' \in B_\epsilon(x)} \ell^{0\text{-}1}((x', y), \mathbf{h}) \right]^1. \tag{1}$$

The adversarial risk of a deterministic classifier $h$ is defined analogously, replacing $\mathbf{h}$ by $h$ in (1).

### 2.2 Threat model

We consider a white-box threat model, in which the adversary has complete access to the classifier proposed by the defender. In theory, we are assuming that the adversary is able to attain (or approximate to any precision) the inner supremum in (1). In practice, if the classifier is differentiable, the adversary will have full access to the gradients (and any derivative of any order), and it will have full knowledge of the classifier, including its architecture and any preprocessing used. The adversary can also query the classifier as many times as needed. The only limitation for the adversary regards the sources of randomness. In particular, when the proposed classifier is probabilistic and an inference pass is done by sampling a label from $\mathbf{h}(x)$, the adversary does not know and cannot manipulate the randomness behind this sampling.

It is worth noting that other works have considered similar inference processes, in which the label of a point is obtained by a sampling procedure [29]. The underlying question is similar to ours: *does randomness improve adversarial robustness?*. They propose a threat model in which an attacker can perform $N$ independent attempts with the goal of producing *at least one misclassification*, which becomes easier if the predicted probability for the correct class is less than 1. In this paper, however, we are interested in the expected accuracy of a classifier from a theoretical point of view, which in practice translates to the probability of error, or the long term average performance or the classifier.

Our formulation of adversarial risk in (1) corresponds to the "constant-in-the-ball" risk in [20] or the "corrupted-instance" risk in [15]. Under this formulation, as pointed out in [40], an adversarial risk of 0 is only possible is the supports of the class-conditioned distributions $\rho_y$ are non-overlapping and separated by at least $2\epsilon$. This is not the case with other formulations of risk introduced in [15, 20]. We focus on the former in this work.

---

[1]The measurability of the 0-1 loss under attack (inner part of (1)) is non-trivial and depends on various factors like the type of ball considered for the supremum (closed or open) [44], and the underlying $\sigma$-algebra [1, 40]. In Section 2.3 and Appendix A we will clarify the assumptions that ensure the well-definedness of the adversarial risk in our setting.

## 2.3 Probabilistic Classifiers Built from a Base Hypothesis Set

In this paper, we study probabilistic classifiers that are constructed from a *base hypothesis set* (BHS) of (possibly infinitely many) deterministic classifiers, denoted by $\mathcal{H}_b$. We use the name *mixtures* [38] for these type of classifiers. In the following, we show that many of the probabilistic classifiers that are commonly used in practice fall under this framework.

Let $\mathcal{H}_b$ be a BHS of deterministic classifiers from $\mathcal{X}$ to $\mathcal{Y}$, which we assume is identifiable with a Borel subset of some $\mathbb{R}^p$. Let $\mu \in \mathcal{P}(\mathcal{H}_b)$ be a probability measure over the measure space $(\mathcal{H}_b, \sigma(\mathcal{H}_b))$, where $\sigma(\mathcal{H}_b)$ denotes the Borel $\sigma$-algebra on $\mathcal{H}_b$. One can construct a probabilistic classifier $\mathbf{h}_\mu : \mathcal{X} \to \mathcal{P}(\mathcal{Y})$, built from $\mathcal{H}_b$, that maps $x \in \mathcal{X}$ to a probability distribution $\mu_x \in \mathcal{P}(\mathcal{Y})$, where $\mu_x(y) = \mathbb{P}_{h \sim \mu}(h(x) = y)$. We now instantiate $\mathcal{H}_b$ and $\mu \in \mathcal{P}(\mathcal{H}_b)$ for two main types of probabilistic classifiers that are commonly used in practice: randomized ensembles and noise injection classifiers.

For a *randomized ensemble classifier (REC)*, $\mu \in \mathcal{P}_M(\mathcal{H}_b) \subset \mathcal{P}(\mathcal{H}_b)$ where $\mathcal{P}_M(\mathcal{H}_b)$ is the set of all discrete measures on $\mathcal{H}_b$ supported on a finite set of $M$ deterministic classifiers, $\{h_1, \ldots, h_M\}$. In this case, $\mathbf{h}_\mu(x)$ takes the value $h_m(x)$ with probability $p_m = \mu(h_m)$ for $m \in [M]$, where $\sum_{m \in [M]} p_m = 1$. RECs were introduced in [38] and play the role of mixed strategies in the adversarial robustness game. They are a simple randomization scheme when a finite number of classifiers are at hand, and both training and attacking them represent a challenge [12, 13].

For a *weight-noise injection classifier (WNC)*, $\mathcal{H}_b = \{h_w : w \in \mathcal{W}\}$ where $h_w$ is a deterministic classifier with parameter $w$. In this case, $\mu$ is taken to be a probability distribution over $\mathcal{W}$ with the understanding that each $w \in \mathcal{W}$ is associated with a unique $h_w \in \mathcal{H}_b$. For example, $\mathcal{H}_b$ can be the set of all neural network classifiers with weights in the set $\mathcal{W} \subseteq \mathbb{R}^p$. Any probability distribution $\mu$ on the space of weights $\mathcal{W}$ results in a probabilistic classifier $\mathbf{h}_\mu$. Alternatively, $\mathcal{H}_b$ can be generated by injecting noise $z$ sampled from a distribution $\mu$ to the weights $w_0$ of a fixed classifier $h_{w_0}$. In this case, the probabilistic classifier $\mathbf{h}_\mu$ maps $x \in \mathcal{X}$ to a probability distribution $\mu_x \in \mathcal{P}(\mathcal{Y})$, where $\mu_x(y) = \mathbb{P}_{z \sim \mu}(h_{w_0+z}(x) = y)$. Weight noise injection has been explicitly used in [22], but there are many other approaches that implicitly define a probability distribution over the parameters of a model [14, 35] and sample one or more at inference.

For an *input-noise injection classifier (INC)*, $\mathcal{H}_b = \{h_\eta : \eta \in \mathcal{X}\}$ where $h_\eta(x) = h_0(x + \eta)$ for a fixed deterministic classifier $h_0$. In this case, $\mu$ is taken to be a probability distribution over $\mathcal{X}$ (which is unrelated to the data generating distribution), and $\mathbf{h}_\mu$ maps $x \in \mathcal{X}$ to a probability distribution $\mu_x \in \mathcal{P}(\mathcal{Y})$, where $\mu_x(y) = \mathbb{P}_{\eta \sim \mu}(h(x + \eta) = y)$. Injecting noise to the input has been used for decades as a regularization method [4, 21]. INCs are studied in [22, 37, 39] as a defense against adversarial attacks, and are closely related to randomized smoothing classifiers [8, 26].

**Remark 1** (Measurability). To ensure the well-definedess of the adversarial risk, we need to ensure that the 0-1 loss under attack (inner part of (1)) is measurable. The 0-1 loss for a fixed class $y \in \mathcal{Y}$ can now be seen as a function from the product space $\mathcal{X} \times \mathcal{H}_b$ to $\{0, 1\}$:

$$f(x, h) = \ell^{0\text{-}1}((x, y), h) = \mathbb{1}\{h(x) \neq y\} \tag{2}$$

For a distribution $\mu \in \mathcal{P}(\mathcal{H}_b)$ and the associated probabilistic classifier $\mathbf{h}_\mu$, we can rewrite the 0-1 loss of $\mathbf{h}_\mu$ at $x$ as

$$\ell^{0\text{-}1}((\cdot, y), \mathbf{h}_\mu) = \mathop{\mathbb{E}}_{h \sim \mu} [\mathbb{1}\{h(\cdot) \neq y\}] = \int_{\mathcal{H}_b} f(\cdot, h) d\mu(h) \tag{3}$$

Note that if $f$ is Borel measurable in $\mathcal{X} \times \mathcal{H}_b$, then by Fubini-Tonelli's Theorem, the 0-1 loss as a function of $x$ with an integral over $\mathcal{H}_b$ shown in (3) is Borel measurable. By [16, Appendix A, Theorem 27], the 0-1 loss under attack is then universally measurable and therefore the adversarial risk, which requires the integral over $\mathcal{X}$ of the loss under attack, is well-defined over this universal $\sigma$-algebra [44, Definition 2.7]. Recent work by Trillos et al. [44] have also shown the existence of Borel measurable solutions for the *closed ball* formulation of adversarial risk.

In this work, we assume that the function $f$ in (2) is Borel measurable in $\mathcal{X} \times \mathcal{H}_b$ for every $y$, so that (1) is well-defined. This assumption is already satisfied in the settings that are of interest for our work. As an example, it holds when the classifiers at hand are neural networks with continuous activation functions. More details on Appendix A. For a deeper study on the measurability and well-definedness of the adversarial risk in different threat models and settings, see [1, 16, 40, 44].

# 3 From Deterministic to Probabilistic Classifiers

Let $\mathcal{H}_b$ be a class of deterministic classifiers. In this section, we compare the robustness of probabilistic classifiers built upon $\mathcal{H}_b$ with the robustness of the class $\mathcal{H}_b$ itself. Note that if we consider the trivial mixtures $\mu = \delta_h$ for $h \in \mathcal{H}_b$, we obtain the original classifiers in $\mathcal{H}_b$, so it is always true that $\inf_{\mu \in \mathcal{P}(\mathcal{H}_b)} \mathcal{R}_\epsilon(\mathbf{h}_\mu) \leq \inf_{h \in \mathcal{H}_b} \mathcal{R}_\epsilon(h)$. We are interested in the situations in which this gap is strict, meaning that mixtures strictly outperform the base classifiers. With this in mind, we introduce the notion of *Matching Penny Gap* to quantify the improvement of probabilistic classifiers over deterministic ones.

**Theoretical Results on Robustness of Probabilistic Classifiers.** We begin by presenting a theorem which shows that the adversarial risk of a probabilistic classifier is at most the average of the adversarial risk of the deterministic classifiers constituting it. The proof of this result will be a first step towards understanding the conditions that favor the mixture classifier.

**Theorem 3.1.** For a probabilistic classifier $\mathbf{h}_\mu : \mathcal{X} \to \mathcal{P}(\mathcal{Y})$ constructed from a BHS $\mathcal{H}_b$ using any $\mu \in \mathcal{P}(\mathcal{H}_b)$, we have $\mathcal{R}_\epsilon(\mathbf{h}_\mu) \leq \mathbb{E}_{h \sim \mu}[\mathcal{R}_\epsilon(h)]$.

A natural follow-up question to is to ask *what conditions guarantee a strictly better performance of a probabilistic classifier*. We know that this gap can be strict, as can be seen in toy examples like the one shown in Figure 1b where $\mathcal{H}_b$ is the set of all linear classifiers, and there exist two distinct classifiers $f_1, f_2$ both attaining the minimum adversarial risk among all classifiers in $\mathcal{H}_b$. Any non-degenerate mixture of $f_1, f_2$ attains a strictly better adversarial risk, demonstrating a strict advantage for probabilistic classifiers.

From the proof of Theorem 3.1 it is clear that a strict gap in performance between probabilistic and deterministic classifiers is only possible when the following strict inequality holds for a non-vanishing probability mass over $(x, y)$.

$$\sup_{x' \in B_\epsilon(x)} \mathbb{E}_{h \sim \mu} \left[ \mathbb{1}\{h(x') \neq y\} \right] < \mathbb{E}_{h \sim \mu} \left[ \sup_{x' \in B_\epsilon(x)} \mathbb{1}\{h(x') \neq y\} \right]. \tag{4}$$

The above condition holds at $(x, y)$ if there exists a subset of vulnerable classifiers $\mathcal{H}_{vb} \subseteq \mathcal{H}_b$ with $\mu(\mathcal{H}_{vb}) > 0$ **any of which can be forced to individually misclassify** the point $(x, y)$ by an adversary using (possibly different) perturbations $x'_h \in B_\epsilon(x)$ for $h \in \mathcal{H}_{vb}$, **but cannot be forced to misclassify all at the same time using the same perturbation** $x' \in B_\epsilon(x)$. Such a configuration is reminiscent of the game of *matching pennies* [18, 46] (see Appendix B). If $\mathcal{H}_b$ is in a *matching penny* configuration at $(x, y)$, a mixed strategy for classification (i.e. a mixture of $\mathcal{H}_b$) achieves a decisive advantage over any pure strategy (i.e. any deterministic base classifier) because the adversary can only force a misclassification on a subset of all vulnerable classifiers. Such a configuration was first noted in [12] in the context of improved adversarial attacks on RECs, and also in [13] for computing the adversarial risk of RECs of size $M = 2$. Figure 1 illustrates such a condition on an example REC of size $M = 2$ over a single point (Figure 1a), and over a simple discrete distribution (Figure 1b). We formalize this intuition with the definition below.

**Definition 1** (Matching penny gap)**.** The matching penny gap of a data point $(x, y) \in (\mathcal{X} \times \mathcal{Y})$ with respect to a probabilistic classifier $\mathbf{h}_\mu$ constructed from $\mathcal{H}_b$ using $\mu \in \mathcal{P}(\mathcal{H}_b)$ is defined as,

$$\pi_{\mathbf{h}_\mu}(x, y) = \mu(\mathcal{H}_{vb}(x, y)) - \mu^{\max}(x, y), \tag{5}$$

where $\mathcal{H}_{vb}(x, y) \subseteq \mathcal{H}_b$ denotes the *vulnerable subset* and $\mu^{\max}(x, y)$ the *maximal simultaneous vulnerability* of base classifiers $\mathcal{H}_b$, defined below.

$$\mathcal{H}_{vb}(x, y) = \{h \in \mathcal{H}_b : \exists x'_h \in B_\epsilon(x) \text{ such that } h(x'_h) \neq y\},$$
$$\mathfrak{H}_{svb}(x, y) = \{\mathcal{H}' \subseteq \mathcal{H}_b : \exists x' \in B_\epsilon(x) \text{ such that } \forall h \in \mathcal{H}', h(x') \neq y\},$$
$$\mu^{\max}(x, y) = \sup_{\mathcal{H}' \in \mathfrak{H}_{svb}(x, y)} \mu(\mathcal{H}').$$

If $\pi_{\mathbf{h}_\mu}(x, y) > 0$, we say that $\mathbf{h}_\mu$ is in *matching penny configuration* at $(x, y)$.

For example, in Figure 1a, $\pi_{\mathbf{h}_\mu}(x_0, y_0) = 1 - \max\{\mu(f_1), \mu(f_2)\} = \min\{\mu(f_1), \mu(f_2)\}$ where $\mathcal{H}_b = \{f_1, f_2\}$. The subset $\mathcal{H}_{vb}(x, y)$ contains all classifiers that can be individually fooled by an

optimal attacker. The collection of subsets $\mathfrak{H}_{svb}(x, y)$ contains all subsets $\mathcal{H}'$ of classifiers that can be *simultaneously fooled*. Then, $\mu^{\max}(x, y)$ is the maximum mass of classifiers that can be fooled simultaneously. Thus, $\pi_{\mathbf{h}_\mu}(x, y)$ measures the gap between the mass of classifiers that are individually vulnerable and the maximum mass of classifiers that can be fooled with only one perturbation.

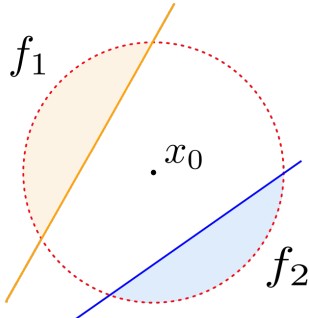

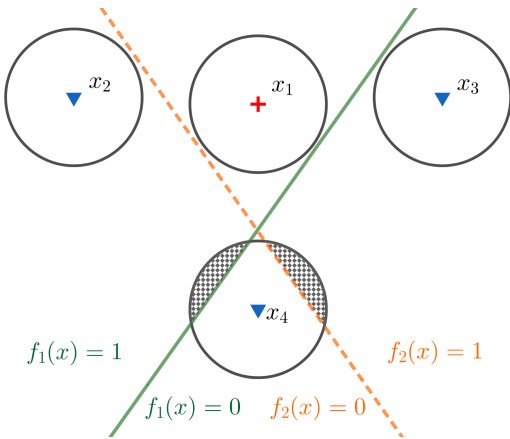

(a) A *matching penny* configuration at $x_0$ with two deterministic classifiers $f_1$ and $f_2$ with $f_1(x_0) = f_2(x_0) = y_0$. In the top-left colored region, $f_1(x) \neq y_0$, and on the bottom-right colored region $f_2(x) \neq y_0$ (vulnerability regions). Both $f_1, f_2$ can be made to incur a loss of 1 at $B_\epsilon(x_0)$ by separate perturbations, but not with the same perturbation.

(b) Binary classification example with discrete data distribution $\frac{1}{2}\delta(x = x_1, y = 1) + \sum_{j=2}^{4} \frac{1}{6}\delta(x = x_j, y = 0)$. No linear classifier can attain adversarial risk better than $\frac{1}{3}$ (attained by both $f_1$ and $f_2$). Any REC with $\mu(f_1) = 1 - \mu(f_2) \in (0, 1)$ achieves a strictly better risk. A REC with $\mu(f_1) = \mu(f_2) = 1/2$ is optimal with risk $\frac{1}{4}$. See Appendix C.4 for details.

Figure 1: Toy examples demonstrating a strict gap in adversarial risk between deterministic and probabilistic classifiers (RECs).

The following theorem strengthens Theorem 3.1 by showing that $\mathcal{R}_\epsilon(\mathbf{h})$ is a strictly convex function if and only if there is a non-zero expected matching penny gap.

**Theorem 3.2.** For a probabilistic classifier $\mathbf{h}_\mu : \mathcal{X} \to \mathcal{P}(\mathcal{Y})$ constructed from a BHS $\mathcal{H}_b$ using any $\mu \in \mathcal{P}(\mathcal{H}_b)$,

$$\mathcal{R}_\epsilon(\mathbf{h}_\mu) = \mathop{\mathbb{E}}_{h \sim \mu}[\mathcal{R}_\epsilon(h)] - \mathop{\mathbb{E}}_{(x,y) \sim \rho}[\pi_{\mathbf{h}_\mu}(x, y)]. \tag{6}$$

*Proof sketch.* By interchanging expectations over $h \sim \mu$ and $(x, y) \sim \rho$ and using the fact that $\sup_{x' \in B_\epsilon(x)} \mathbb{1}\{h(x') \neq y\} = 1$ if and only if $h \in \mathcal{H}_{vb}(x, y)$, we first show the following.

$$\mathop{\mathbb{E}}_{h \sim \mu}[\mathcal{R}_\epsilon(h)] = \mathop{\mathbb{E}}_{(x,y) \sim \rho}[\mu(\mathcal{H}_{vb}(x, y))]. \tag{7}$$

Arguing from the definition of $\mu^{\max}(x, y)$, we then show that $\sup_{x' \in B_\epsilon(x)} \mathbb{E}_{h \sim \mu}[\mathbb{1}\{h(x') \neq y\}] = \mu^{\max}(x, y)$. Taking expectation with respect to $(x, y) \sim \rho$, we get,

$$\mathcal{R}_\epsilon(\mathbf{h}_\mu) = \mathop{\mathbb{E}}_{(x,y) \sim \rho}[\mu^{\max}(x, y)]. \tag{8}$$

Combining (8) and (7) yields (6). □

The following corollary shows that a lower bound on the expected matching penny gap is both necessary and sufficient for a mixture to strictly outperform any deterministic classifier in $\mathcal{H}_b$.

**Corollary 3.1.** For $\mu' \in \mathcal{P}(\mathcal{H}_b)$, $\mathcal{R}_\epsilon(\mathbf{h}_{\mu'}) < \inf_{h \in \mathcal{H}_b} \mathcal{R}_\epsilon(h)$ if and only if the following condition holds.

$$\mathop{\mathbb{E}}_{(x,y) \sim \rho}[\pi_{\mathbf{h}_{\mu'}}(x, y)] > \mathop{\mathbb{E}}_{h \sim \mu'}[\mathcal{R}_\epsilon(h)] - \inf_{h \in \mathcal{H}_b} \mathcal{R}_\epsilon(h) \tag{9}$$

Additionally, $\inf_{\mu \in \mathcal{P}(\mathcal{H}_b)} \mathcal{R}_\epsilon(\mathbf{h}_\mu) < \inf_{h \in \mathcal{H}_b} \mathcal{R}_\epsilon(h)$ if and only if there exists $\mu' \in \mathcal{P}(\mathcal{H}_b)$ for which (9) holds.

**Remark 2** (Multiple optimal classifiers in matching penny configuration). For finite $\mathcal{H}_b$, the assumption (9) of Corollary 3.1 holds whenever there exist two distinct optimal classifiers $h_1^*, h_2^* \in \mathcal{H}_b$ with a positive expected matching penny gap. In such a case, the RHS of (9) is zero for any $\mu$ that is a mixture of $h_1^*, h_2^*$. This is indeed the case in Figure 1b. More generally, assumption (9) holds if there is a subset of classifiers in a matching penny configuration that are "near optimal" on average. More details in Appendix C.3.

The following example depicts the scenario of Remark 2 in the extreme case in which there exist infinitely many distinct optimal classifiers in matching penny configuration, leading to the maximum possible advantage of using probabilistic classifiers.

**Example 1** (Maximum expected matching penny gap of 1). Consider a discrete data distribution $\rho = \delta(x = 0, y = 1)$ and a BHS composed of infinitely many linear classifiers, $\mathcal{H}_b = \{h : h(x) = \mathbb{1}\{w^T x < 1\}, \|w\|_2 = \frac{1}{\epsilon}\}$ where $\|\cdot\|_2$ is the Euclidean norm. Observe that every classifier in $\mathcal{H}_b$ is vulnerable at $(x, y) = (0, 1)$ and so $\mathcal{R}_\epsilon(h) = 1$ for all $h \in \mathcal{H}_b$. However, no pair is simultaneously vulnerable. Hence, $\pi_{\mathbf{h}_\mu}(x, y) = 1$ for any distribution $\mu \in \mathcal{P}(\mathcal{H}_b)$ that is continuous over the entire support $\mathcal{H}_b$. By Theorem 3.2 we get that $\mathcal{R}_\epsilon(\mathbf{h}_\mu) = 0$ for any such $\mu$. Therefore, any such $\mathbf{h}_\mu$ outperforms every $h \in \mathcal{H}_b$, and we have $0 = \inf_{\mu \in \mathcal{P}(\mathcal{H}_b)} \mathcal{R}_\epsilon(\mathbf{h}_\mu) < \inf_{h \in \mathcal{H}_b} \mathcal{R}_\epsilon(h) = 1$. More details on Appendix C.1.

**Remark 3** (Probabilistic classifiers with zero matching penny gap are not useful). Suppose the expected matching penny gap $\mathbb{E}_{(x,y) \sim \rho}[\pi_{\mathbf{h}_{\mu'}}(x, y)]$ is zero for some $\mu' \in \mathcal{P}(\mathcal{H}_b)$. Then the left hand side of (9) is zero, whereas the right-hand side $\mathbb{E}_{h \sim \mu'}[\mathcal{R}_\epsilon(h)] - \inf_{h \in \mathcal{H}_b} \mathcal{R}_\epsilon(h)$ is always non-negative. Hence, (9) does not hold and so $\mathcal{R}_\epsilon(\mathbf{h}_{\mu'}) \geq \inf_{h \in \mathcal{H}_b} \mathcal{R}_\epsilon(h)$. Therefore, any such probabilistic classifier underperforms the best deterministic classifier in the base set $\mathcal{H}_b$.

The following example illustrates a scenario described in Remark 3, where we examine classifiers with decision boundaries that are parallel sets [23].

**Example 2** (Minimum expected matching penny gap of 0 / Parallel decision boundaries). Fix any binary classifier $h : \mathcal{X} \to \{0, 1\}$ with non-empty decision region $A_h = \{x \in \mathcal{X} : h(x) = 1\} \subset \mathcal{X} \subseteq \mathbb{R}^d$. Let $\mathcal{H}_b$ be composed of all classifiers whose decision regions are $r$-parallel sets of $A_h$ defined as $A_h^r = A_h \oplus B_r(0)$ where $\oplus$ denotes the Minkowski sum, i.e., $\mathcal{H}_b = \{h : \exists r \geq 0 \text{ s.t. } A_h^r = \{x \in \mathcal{X} : h(x) = 1\}\}$. Because of the parallel decision boundaries, whenever two classifiers in $\mathcal{H}_b$ are vulnerable, they are simultaneously vulnerable, and never exhibit a matching penny configuration. Therefore, $\mathbb{E}_{(x,y) \sim \rho}[\pi_{\mathbf{h}_\mu}(x, y)] = 0$ for any $\mu \in \mathcal{P}(\mathcal{H}_b)$. More details on Appendix C.2.

**Application to RECs and Links with [13].** In the case of RECs where $\mu = \sum_{m \in [M]} p_m \delta_{h_m}$ i.e., $\mathbf{h}_\mu$ is a mixture of $M$ deterministic classifiers in $\mathcal{H}_b = \{h_m\}_{m \in [M]}$, we can instantiate Theorem 3.2 as follows. As the family $\mathfrak{H}_{svb}(x, y)$ is finite, the supremum $\mu^{\max}(x, y)$ is always attained by some subset of simultaneously vulnerable classifiers $\mathcal{H}_{svb}^{max}(x, y)$. We can then write:

$$\mathcal{R}_\epsilon(\mathbf{h}_\mu) = \sum_{m \in [M]} p_m \mathcal{R}_\epsilon(h_m) - p_m \left[ \mathbb{E}_{(x,y) \sim \rho} \mathbb{1}\{h_m \in \mathcal{H}_{vb}(x, y) \setminus \mathcal{H}_{svb}^{max}(x, y)\} \right] \quad (10)$$

Alternatively, we can use (8) to write $\mathcal{R}_\epsilon(\mathbf{h}_\mu) = \sum_{m \in [M]} p_m \mathbb{E}_{(x,y) \sim \rho} \mathbb{1}\{h_m \in \mathcal{H}_{svb}^{max}(x, y)\}$. At each $(x, y)$, testing whether $h_m \in \mathcal{H}_{svb}^{max}(x, y)$ reduces to solving a combinatorial optimization problem, as noted in [13]. Any $(x, y)$ falls into one of finitely many configurations, depending on which subset of classifiers are vulnerable or simultaneously vulnerable at $(x, y)$. Dbouk and Shanbhag [13] use this to derive upper and lower bounds on $\mathcal{R}_\epsilon(\mathbf{h}_\mu)$. Specifically, [13, Proposition 1] is equivalent to (8) for the special case of RECs. Also, [13, Theorem 1] can be proved as an application of Theorem 3.2 to RECs with $M = 2$. To establish the link, one must note that $\mathbb{E}_{(x,y) \sim \rho}[\pi_{\mathbf{h}_{\mu'}}(x, y)] = \frac{1}{2}\rho(\{(x, y) \in R\})$ where $R \subseteq \mathcal{X} \times \mathcal{Y}$ indicates the set of all points where $h_1, h_2$ are in matching penny configuration.

## 4 From Probabilistic to Deterministic Classifiers

In this section, we prove that for any probabilistic binary classifier $\mathbf{h}$, there exists a deterministic classifier $h$ in a "threshold" hypothesis set $\mathcal{H}_T(\mathbf{h})$ with at least the same adversarial risk. In

Section 4.1 we present the main theorem and in Section 4.2 we apply the theorem to various classes of probabilistic classifiers.

## 4.1   Reducing a Probabilistic Classifier to a Deterministic One Through Threshold Classifiers

In the case of binary classification, i.e. $\mathcal{Y} = \{0, 1\}$ any distribution $\nu \in \mathcal{P}(\mathcal{Y})$ is uniquely determined by a scalar $\alpha = \nu(y = 1) \in [0, 1]$. Hence, any probabilistic binary classifier is identified with a function $\mathbf{h} : \mathcal{X} \to [0, 1]$, where $\mathbf{h}(x) = \nu(y = 1 | x) \in [0, 1]$. Accordingly, $\ell^{0\text{-}1}((x, 0), \mathbf{h}) = \mathbf{h}(x)$ and $\ell^{0\text{-}1}((x, 1), \mathbf{h}) = 1 - \mathbf{h}(x)$.

Given a probabilistic binary classifier $\mathbf{h} : \mathcal{X} \to [0, 1]$ and a threshold $\alpha \in [0, 1]$, the $\alpha$-*threshold classifier* $h^\alpha : \mathcal{X} \to \{0, 1\}$ is defined as $h^\alpha(x) = \mathbb{1}\{\mathbf{h}(x) > \alpha\}$, and the *threshold hypothesis set (THS)* of $\mathbf{h}$ is given by $\mathcal{H}_T(\mathbf{h}) = \{h^\alpha\}_{\alpha \in [0,1]}$. In Theorem 4.1 we show that there exists $h^{\alpha_*} \in \mathcal{H}_T(\mathbf{h})$ such that $\mathcal{R}_\epsilon(h^{\alpha_*}) \leq \mathcal{R}_\epsilon(\mathbf{h})$. The following lemma plays a crucial role in proving Theorem 4.1.

**Lemma 4.1.** Let $\mathbf{h} : \mathcal{X} \to [0, 1]$ be any measurable function. For any $\succ \in \{>, \geq\}$, the following inequality holds, and it becomes an equality if $\mathbf{h}$ is continuous or takes finitely many values:

$$\mathbb{1}\left\{ \left( \sup_{x' \in B_\epsilon(x)} \mathbf{h}(x') \right) \succ \alpha \right\} \geq \sup_{x' \in B_\epsilon(x)} \mathbb{1}\{\mathbf{h}(x') \succ \alpha\} \tag{11}$$

Note that Lemma 4.1 is a generalization of the layer-cake representation of $\mathbf{h}(x)$ given by,

$$\mathbf{h}(x) = \int_0^1 \mathbb{1}\{\mathbf{h}(x) > \alpha\}d\alpha = \int_0^1 \mathbb{1}\{\mathbf{h}(x) \geq \alpha\}d\alpha.$$

**Theorem 4.1.** Let $\mathbf{h} : \mathcal{X} \to [0, 1]$ be any probabilistic binary classifier. Let $h^\alpha$ be the $\alpha$-*threshold* classifier. Then the following equation holds:

$$\mathcal{R}_\epsilon(\mathbf{h}) \geq \int_0^1 \mathcal{R}_\epsilon(h^\alpha)d\alpha \geq \inf_\alpha \mathcal{R}_\epsilon(h^\alpha). \tag{12}$$

Further, if $\mathbf{h}$ is either continuous or takes finitely many values, the first inequality in (12) becomes an equality.

Note that $\mathbf{h}$ takes finitely many values in the case of RECs, and $\mathbf{h}$ is continuous in the case of INCs and WNCs whenever the noise distribution admits a density. Hence, $R_\epsilon(\mathbf{h}) = \int_0^1 \mathcal{R}_\epsilon(h^\alpha)d\alpha$ in all these cases.

**Remark 4.** Theorem 4.1 says that in the binary case (K = 2), if one is able to consider complex enough hypotheses sets that contain the $\mathcal{H}_T(\mathbf{h})$, then randomization is not necessary because there is a deterministic classifier with equal or better adversarial risk.

It was very recently shown with a toy example [45, Section 5.2] that Theorem 4.1 does not hold in the multi-class case $K > 2$. This example shows that even when the family of probabilistic classifiers considered is very general (all Borel measurable functions), simple data distributions can create a situation in which the optimal classifier is probabilistic, and there is no optimal deterministic classifier. In other words, there is a strict gap in adversarial risk between the best deterministic classifier and the best probabilistic one.

## 4.2   Applications: Connections to Weighted Ensembles and Randomized Smoothing

In this section, we apply Theorem 4.1 to probabilistic classifiers presented in Section 2.3.

**RECs.** When $\mu = \sum_{m \in [M]} p_m \delta_{h_m}$ and $\mathcal{Y} = \{0, 1\}$, the REC $\mathbf{h}_\mu$ can be written as $\mathbf{h}_\mu(x) = \sum_{m=1}^{M} p_m h_m(x)$. Let us introduce the constant classifier $h_0(x) = 1$ for all $x$, and $p_0 = -\alpha$. Then $h^\alpha(x) = \mathbb{1}\{\sum_{m=1}^{K} p_m h_m(x) > \alpha\} = \mathbb{1}\{\sum_{m=0}^{M} p_m h_m(x) > 0\}$. This shows that $h^\alpha$ is a *weighted ensemble*, such as those that the boosting algorithm can learn [17].

Further, a REC $\mathbf{h}$ built with $M$ base binary classifiers can take at most $p \leq 2^M$ distinct values, corresponding to all possible partial sums of the weights $p_m$. Let $0 = r_1 \leq \ldots \leq r_p = 1$ be

the possible values. Then, for any $\alpha \in [r_i, r_{i+1})$, $h^\alpha = h^{r_i}$. Applying Theorem 4.1 yields,

$$\mathcal{R}_\epsilon(\mathbf{h}) = \int_0^1 \mathcal{R}_\epsilon(h^\alpha) d\alpha = \sum_{i=1}^{p-1} \int_{r_i}^{r_{i+1}} \mathcal{R}_\epsilon(h^{r_i}) d\alpha = \sum_{i=1}^{p-1} (r_{i+1} - r_i) \mathcal{R}_\epsilon(h^{r_i}) \tag{13}$$

Equation (13) shows that the THS $\mathcal{H}_T(\mathbf{h}) = \{h^\alpha\}_{\alpha \in [0,1]}$ is composed of at most $p \leq 2^M$ distinct classifiers, each of which is in turn a weighted ensemble. In case of uniform weights i.e. $p_m = \frac{1}{M}$, there are only $M$ distinct weighted ensembles in $\mathcal{H}_T(\mathbf{h})$.

**INCs.** Let $h_\eta : \mathcal{X} \rightarrow \{0,1\}$ be the deterministic binary classifiers created from a base classifier $h$, as defined in Section 2.3. The probabilistic classifier $\mathbf{h}_\mu$ is defined as $\mathbf{h}_\mu(x) = \mathbb{P}_{\eta \sim \mu}(h(x + \eta) = 1)$. Thus, the $\alpha$-threshold classifier takes a similar form to the well-known classifier obtained by *randomized smoothing*.

$$h^\alpha(x) = \mathbb{1}\{ \mathbb{P}_{\eta \sim \mu} (h(x + \eta) = 1) > \alpha\}. \tag{14}$$

Randomized smoothing was first introduced in [26] and refined in [8, 27, 42] as a way to create classifiers that are certifiably robust. Given a deterministic classifier $h : \mathcal{X} \rightarrow \mathcal{Y}$ and a noise distribution $\mu \in \mathcal{P}(\mathcal{X})$, the smoothed model will output a prediction according to the following rule:

$$h_{RS(\mu)}(x) = \underset{y \in \mathcal{Y}}{\mathrm{argmax}} \, \mathbb{P}_{\eta \sim \mu} (h(x + \eta) = y) = \mathbb{1} \left\{ \mathbb{P}_{\eta \sim \mu} (h(x + \eta) = 1) > \tfrac{1}{2} \right\} \tag{15}$$

In other words, Equation (14) generalizes the randomized smoothing classifier in the binary case by replacing the $\frac{1}{2}$ threshold by $\alpha$. Theorem 4.1 then states that *for any INC, there exists a deterministic randomized smoothing classifier with threshold $\alpha$ that is at least as robust.*

## 5  Families of Binary Classifiers for Which Randomization Does not Help

In Sections 3 and 4, we discussed two types of hypothesis sets for binary classification: 1) base set $\mathcal{H}_b$ from which a probabilistic classifier is built using some $\mu \in \mathcal{P}(\mathcal{H}_b)$, and 2) threshold set $\mathcal{H}_T(\mathbf{h})$ which is built from a base probabilistic classifier $\mathbf{h}$. Recapitulating this two-step process, one may define the *completion* of a base set $\mathcal{H}_b$ of binary classifiers w.r.t. a set of probability distributions $\mathcal{M} \subseteq \mathcal{P}(\mathcal{H}_b)$ as, $\overline{\mathcal{H}_b}(\mathcal{M}) = \cup_{\mu \in \mathcal{M}} \mathcal{H}_T(\mathbf{h}_\mu)$. Observe that $\mathcal{H}_b \subseteq \overline{\mathcal{H}_b}(\mathcal{P}(\mathcal{H}_b))$. In the following theorem, we show that if $\mathcal{H}_b = \overline{\mathcal{H}_b}(\mathcal{M})$, then probabilistic binary classifiers built from $\mathcal{H}_b$ using any $\mu \in \mathcal{M}$ do not offer robustness gains compared to the deterministic classifiers in $\mathcal{H}_b$.

**Theorem 5.1.** If $\mathcal{H}_b = \overline{\mathcal{H}_b}(\mathcal{M})$ and $\delta_h \in \mathcal{M}$ for all $h \in \mathcal{H}_b$, then $\inf_{h \in \mathcal{H}_b} \mathcal{R}_\epsilon(h) = \inf_{\mu \in \mathcal{M}} \mathcal{R}_\epsilon(\mathbf{h}_\mu)$.

*Proof.*

$$\inf_{h \in \overline{\mathcal{H}_b}(\mathcal{M})} \mathcal{R}_\epsilon(h) = \inf_{\mu \in \mathcal{M}} \inf_{h \in \mathcal{H}_T(\mathbf{h}_\mu)} \mathcal{R}_\epsilon(h) \leq \inf_{\mu \in \mathcal{M}} \mathcal{R}_\epsilon(\mathbf{h}_\mu) \leq \inf_{h \in \mathcal{H}_b} \mathcal{R}_\epsilon(h), \tag{16}$$

where the first inequality follows from Theorem 4.1 and the second by considering $\mu = \delta_h$ for any $h \in \mathcal{H}_b$. The desired conclusion follows by observing that the first and last terms in (16) are identical from the assumption $\mathcal{H}_b = \overline{\mathcal{H}_b}(\mathcal{M})$. $\square$

A trivial example where the assumptions of Theorem 5.1 hold is when $\mathcal{H}_b$ is the set of all measurable functions $h : \mathcal{X} \rightarrow \{0,1\}$. Such a $\mathcal{H}_b$ is commonly used in the study of optimal adversarial risk [2, 40]. In the following corollary, we show that the assumptions of Theorem 5.1 also hold in the case of RECs built using $\mathcal{H}_b$ that satisfy a "closure" property.

**Corollary 5.1.** Let $\mathcal{H}_b$ be any family of deterministic binary classifiers. Let $\mathcal{M} = \mathcal{P}_M(\mathcal{H}_b) \subset \mathcal{P}(\mathcal{H}_b)$ be the subset of probability measures over $\mathcal{H}_b$ defining RECs as in Section 2.3. Let $\mathcal{A} = \left\{ h^{-1}(1) : h \in \mathcal{H}_b \right\}$. If $\mathcal{A}$ is closed under union and intersection, then

$$\inf_{h \in \mathcal{H}_b} \mathcal{R}_\epsilon(h) = \inf_{\mu \in \mathcal{P}_M(\mathcal{H}_b)} \mathcal{R}_\epsilon(\mathbf{h}_\mu).$$

**Remark 5** (Implications on the optimal adversarial classifier). Pydi and Jog [40, Theorem 8] show the existence of a binary deterministic classifier attaining the minimum adversarial risk over the space of all Lebesgue measurable classifiers in $\mathbb{R}^d$. Awasthi et al. [2, Theorem 1] prove a similar result over the space of universally measurable classifiers in $\mathbb{R}^d$. For both these families the assumptions of Theorem 5.1 hold for $\mathcal{M} = \mathcal{P}(\mathcal{H}_b)$, and so probabilistic classifiers offer no additional advantage over deterministic ones. Further, suppose $\mathcal{H}^*$ is a finite subset of classifiers achieving optimal adversarial risk. Then by Theorem 3.2, for any $\mu \in \mathcal{P}(\mathcal{H}^*)$, the expected matching penny gap for $\mathbf{h}_\mu$ must be zero, otherwise we could strictly improve the adversarial risk by considering the mixture built using $\mu$. In other words, any pair of optimally robust binary deterministic classifiers in the settings of [2, 40] can only be in a matching penny configuration on a null subset $E \subset \mathcal{X} \times \mathcal{Y}$.

# 6 Conclusion and Future Work

In this paper, we have studied the robustness improvements brought by randomization. First, we studied the situation in which one expands a base family of classifiers by considering probability distributions over it. We showed that under some conditions on the data distribution and the configuration of the base classifiers, such a probabilistic expansion could offer gains in adversarial robustness (See Corollary 3.1), characterized by the *matching penny gap*. These results generalize previous work that focused on RECs [13]. Next, we showed that for any binary probabilistic classifier, there is always another deterministic extension with classifiers of comparable or better robustness. This result is linked with the existence results in [2, 40]. As a direct consequence of this result, we show that in the binary setting, deterministic weighted ensembles are at least as robust as randomized ensembles and randomized smoothing is at least as robust as noise injection.

There are many avenues for future research.

**Improving the Matching Penny Gap.** An interesting direction would be finding tighter bounds on the matching penny gap for particular and widely used probabilistic classifiers like RECs, INCs and WNCs. It would be interesting to establish the conditions under which each method can offer robustness gains, and to quantify those gains in terms of parameters such as the strength of the noise injected or the number of classifiers in the REC.

**Studying the Threshold Hypothesis Sets.** We have seen in Section 4.2 that different probabilistic classifiers induce different THS. In particular, we showed the THS corresponding to RECs and INCs. It would be useful to formally describe the THS induced by other popular families of probabilistic classifiers in the literature. It would also be useful to quantify the complexity gap between the initial $\mathcal{H}_b$ and the THS to understand the risk-complexity trade-off we would have to incur.

**Multiclass Setting.** The toy example in [45, Section 5.2] shows that randomization might be necessary in the multi-class setting, as it is no longer true that there is always a deterministic optimal classifier, even when considering very general families of deterministic classifiers like all measurable functions. A natural road for future work is to further characterize the situations in which probabilistic classifiers strictly outperform deterministic ones in the multi-class setting.

**Training algorithms for diverse ensembles and RECs.** There are numerous works based on the idea of training diverse classifiers that are not simultaneously vulnerable to create robust ensembles [11, 24, 33, 41, 49, 50]. Most of these approaches try to induce orthogonality in decision boundaries so that gradient-based attacks do not transfer between models. This intuition is validated by Corollary 3.1, since such diversification would tend to increase the matching penny gap. It should be noted that ensembles and RECs have different inference procedures, and attacking them represents different optimization problems. One avenue of research would be to make the link between the matching penny gap and the diversity more formal. In addition, designing training algorithms explicitly optimizing the matching penny gap would be valuable, particularly because existing training algorithms for RECs [31, 38] have been shown to be inadequate [12].

# 7 Acknowledgements

This work was funded by the French National Research Agency (DELCO ANR-19-CE23-0016).

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

# Supplementary Material

## Appendix A  Measurability of the $\ell^{0\text{-}1}$ under attack

In this section, we will go over the conditions that ensure the well-definedness of the adversarial risk defined in (1). We will show that the assumptions we make are verified in the settings that are of interest to our work. We highlight that the main purpose of this paper is not to deeply study the existence of measurable solutions to the adversarial problem, as was done in previous work [1, 40, 44]. In this work, we allow ourselves to add hypothesis on the input space $\mathcal{X}$ and the families of classifiers $\mathcal{H}_b$ that we consider, as long as they are valid in practice.

### A.1  Discussion

Recall that in our setting we have the following. The input space $\mathcal{X}$ is some subset (almost always bounded or even compact) of $\mathbb{R}^d$, $\mathcal{Y} = [K]$ and $\mathcal{P}(\mathcal{Y}) = \Delta^K$. On the other hand, for all the applications of interest to our work, we consider *base hypothesis sets* $\mathcal{H}_b$ that are either finite or discrete (for randomized ensembles), or identifiable with subsets of some $\mathbb{R}^p$ (for weight and input noise injection). For all these sets, we assume the usual topology and the Borel $\sigma$-algebra that is generated by the open sets. We further assume that $\mathcal{X}$ and $\mathcal{H}_b$ **are Borel spaces**, which is not restrictive. It allows, for example, any open or closed set of $\mathbb{R}^n$, which already encompasses all practical applications of interest. This hypothesis makes it simpler to work on the product measure space $\mathcal{X} \times \mathcal{H}_b$.

In the adversarial attacks literature, the question of measurability is always with respect to $\mathcal{X}$ [2, 40]. For a deterministic classifier $h : \mathcal{X} \to \mathcal{Y} = [K]$ to be measurable, the sets $h^{-1}(k)$ must be measurable. In other words, each classifier $h$ creates a $K$-partition of $\mathcal{X}$ into measurable subsets $h^{-1}(1), \ldots, h^{-1}(K)$. This immediately translates into the $\ell^{0\text{-}1}$, as a function of $x$, being measurable, because $\ell^{0\text{-}1}((\cdot, y), h)^{-1}(1) = \{x \in \mathcal{X} \mid h(x) \neq y\} = \mathcal{X} \setminus h^{-1}(y)$. In other words, assuming classifiers are measurable functions translates into measurability of the 0-1 loss, so the arguments in previous work [1, 16, 40] can be used to justify that the adversarial risk is well-defined.

We introduce a new component, the BHS $\mathcal{H}_b$, and we now compute integrals over it. For this reason, we now consider more general functions of the form $f : \mathcal{X} \times \mathcal{H}_b \to \{0, 1\}$, $f(x, h) = \mathbb{1}\{h(x) \neq y\}$ and study their measurability. If one fixes $h$, the function $f^h(x) = \mathbb{1}\{h(x) \neq y\}$ becomes the same 0-1 loss considered earlier, so assuming every $h$ is a measurable function translates into the fact that $f$ is measurable w.r.t. $x$ for every fixed $h$. The new part we have to deal with is the measurability of $f$ w.r.t $h$ for every fixed $x$.

As a first example, let us consider the case of linear classifiers. In this case, a classifier $h$ can be represented as a $K \times d$ matrix $A$ with the classification rule $h(x) = \mathrm{argmax}_k (Ax)_k$, meaning $h$ classifies $x$ using the row with higher score. In this case, we can directly see the measurability of $f$ in each component ($f^h$ and $f^x$) by looking at the pre-images of 1, which are the sets that induce an error (recall $y$ is fixed). For $f^h$, the pre-image is the set of $x \in \mathcal{X}$ such that $h(x) \neq y$. This can be seen as the complement of the set $X_y$ in which $h$ predicts $y$.

$$(f^h)^{-1}(1) = \{x \in \mathcal{X} \mid h(x) \neq y\} = \{x \in \mathcal{X} \mid h(x) = y\}^C = X_y^C$$

The set $X_y$ is described by $K - 1$ linear equations using the rows $a_j$ of the matrix $A$ that defines the linear classifier $h$, as follows

$$X_y = \{x \in \mathcal{X} \mid a_y^T x \geq a_j^T x, \ \forall j \neq y\}$$

This set is the finite intersection of hyperplanes, so it is Borel measurable, and therefore the function $f^h$ is measurable on $\mathcal{X}$ for any $h$ linear classifier.

For the measurability of $f^x$, we have to consider the set $H_y$ of classifiers that produce a prediction of $y$ for a fixed $x$. Let us recall that each classifier is represented by a matrix $A \in \mathbb{R}^{K \times d}$, then

$$H_y = \{h \in \mathcal{H}_b \mid h(x) = y\} = \{A \in \mathbb{R}^{K \times d} \mid a_y^T x \geq a_j^T x, \ \forall j \neq y\}$$

$H_y$ is a convex cone, because if $A, B \in H_y$, then for a positive $\alpha$, $\alpha A \in H_y$ and $A + B \in H_y$. This set is therefore measurable, and so we can conclude that $f^x$ is measurable for every $x$.

When considering neural networks with weights parametrized by $w$, we can think of them as non-linear functions plus a last linear layer as the one described earlier. Then, if the non-linear function is continuous w.r.t both $x$ and $w$, the function $f$ we are considering is again measurable w.r.t to $x$ and $w$.

In summary, all the situations that are of interest to our work verify that the function $f$ is defined over a product space $\mathbb{R}^d \times \mathbb{R}^p$, and it has good properties when seen as functions of $x$ or $h$ separately (measurable, continuous or differentiable). These *separately measurable functions* [7, Definition 4.47] might not be *jointly measurable*, *i.e.* measurable on the product space, which is one condition that would grant us the well-definedess of (1). Nevertheless, we can make use of the stronger properties these functions have and assume they are *Carathéodory functions* [7, Definition 4.50]. This will imply the joint measurability of $f$. For completeness, we rewrite the results from [7]:

**Definition 2** (Carahéodory function [7, Definition 4.50])**.** Let $(S, \Sigma)$ be a measurable space, and let $X$ and $Y$ be topological spaces. Let $\mathcal{B}_Y$ be the Borel $\sigma$-algebra on $Y$. A function $f : S \times X \to Y$ is a **Carathéodory function** if:

- For each $x \in X$, the function $f^x = f(\cdot, x) : S \to Y$ is $(\Sigma, \mathcal{B}_Y)$-measurable.

- For each $s \in S$, the function $f^s = f(s, \cdot) : X \to Y$ is continuous.

As we have seen, the functions $f : \mathcal{X} \times \mathcal{H}_b \to \mathbb{R}$ we consider can be assumed to be Carthéodory functions, as they are in general differentiable or continuous in both components. Even going away from practice, in theoretical works it is usual to consider classifiers $h : \mathcal{X} \to \Delta^K$ as Borel measurable functions from $\mathcal{X}$, and adding the assumption of continuity w.r.t $h$ for every fixed $x$ it not restrictive (for example neural networks with continuous activation functions [31]).

**Lemma A.1** (Carahéodory functions are jointly measurable [7, Lemma 4.51])**.** Let $(S, \Sigma)$ be a measurable space, $X$ a separable metrizable space, and $Y$ a metrizable space. Then every Carathéodory function $f : S \times X \to Y$ is jointly measurable

## A.2 Conclusion

If we assume the hypotheses of Lemma A.1, that is

1. We assume that for every component $y$, the function $f : \mathcal{X} \times \mathcal{H}_b \to \{0, 1\}$ defined as $f(x, h) = \mathbb{1}\{h(x) \neq y\}$ is a *Carathéodory function*, which translates in our case into:
   - Each $h \in \mathcal{H}_b$, $h$ is a measurable function on $\mathcal{X}$ (very common assumption).
   - For each $x$, the function $f^x : \mathcal{H}_b \to \{0, 1\}$, $f^x(h) = f(x, h)$ is continuous (not unrealistic, satisfied by neural networks with continuous activation functions)
2. The input space $\mathcal{X}$ is a measurable space (satisfied by hypothesis, $\mathcal{X} \subseteq \mathbb{R}^d$ is some Borel set).
3. The base hypothesis set $\mathcal{H}_b$ is a separable metrizable space (valid when $\mathcal{H}_b$ is a Borel subset of $\mathbb{R}^p$).
4. The output space $\{0, 1\}$ is a metrizable space (satisfied).

then, by Lemma A.1, the function $f$ is jointly measurable.

By Fubini-Tonelli's Theorem, the function $x \to \int_{\mathcal{H}_b} f(x, h) d\mu(h)$ is measurable on $\mathcal{X}$. Then, following the same arguments as in [1, 16], the loss under attack can be defined and is measurable over the universal $\sigma$-algebra $\mathcal{U}(\mathcal{X})$ on $\mathcal{X}$. Then, considering $\mathcal{U}(\mathcal{X})$, the adversarial risk is well-defined as it is an integral over $\mathcal{X}$ using the completions of the class conditioned measures $\rho_y$ over $\mathcal{U}(\mathcal{X})$ (see (1)).

## Appendix B   On the name *Matching penny gap*

In the original matching penny game between player 1 (`attacker`) and player 2 (`defender`), each player has a penny coin and has to secretly position its penny in either *heads* or *tails*. Then, both coins are simultaneously revealed and the outcome of the game is decided as follows: `attacker` wins if both pennies match. If they do not match, then `defender` wins. This situation can be represented using the following matrix (`attacker` wants to maximize), where *heads* and *tails* have been replaced by $f_1$ and $f_2$ for reasons that will be explained later:

|  |  | Attacker | |
|---|---|---|---|
|  |  | $f_1$ | $f_2$ |
| **Defender** | $f_1$ | 1 | 0 |
|  | $f_2$ | 0 | 1 |

This game has no Pure Nash Equilibrium. It has, however, a mixed Nash equilibrium which consist of the strategies $(\frac{1}{2}, \frac{1}{2})$ for both players [18]. In the context of the *matching pennies game*, this can be seen as the strategy in which players toss the coin to obtain either *heads* or *tails* uniformly at random (assuming the coins are fair) instead of choosing one of the sides. Using this strategy, each player can be sure that, on average over multiple realizations of the game, they will win half of the times each. The fact that it is a Nash Equilibrium means that, if one player deviated from this strategy, the other player could find a way to win more than half the times. To see this, imagine that the defender player plays *heads* with probability $\frac{1}{2} + \delta$. Then, if attacker knew this, he could play the strategy *always heads*. Then, from all the realizations of the game, with probability $\frac{1}{2} + \delta$ the coins would be both on the same side (*heads*), meaning attacker would win $\frac{1}{2} + \delta$ of the times, gaining $\delta$ w.r.t the equilibrium strategy.

The parallel with our work can be simply explained as follows (see Figure 2): Take a point $(x_0, y_0)$ and two classifiers $f_1, f_2$ that correctly classify $x_0$. Suppose that **both $f_1$ and $f_2$ are vulnerable at $x_0$**, which is represented by the colored areas in Figure 2. However, add the key assumption that **they cannot be attacked at the same time**, which is represented by the fact that the colored areas do not intersect inside the $\epsilon$-ball around $x_0$. That is, even though an optimal attacker can fool each

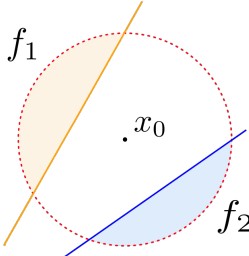

Figure 2: Example of classifiers in a matching penny configuration around a point $x_0$.

classifier individually, there is no point in the allowed perturbation region $B_\epsilon(x_0)$ in which both are simultaneously fooled.

Consider now that the defender is using a randomized ensemble that picks either $f_1$ or $f_2$ at random for inference. This was introduced in [38] with the name mixtures of classifiers, related to mixed strategies in the context of game theory. In such setting, the optimal attacker that faces such mixture is now in a matching pennies game situation. At each inference pass, the attacker must choose which classifier to attack in order to craft the adversarial example $x'$, and if the chosen classifier matches with the one the defender used, then the attacker wins. If they do not match, the prediction made by the defender will be correct and the attacker would have lost. This is exactly the game of *matching pennies*!

What is particularly worse for defender is that, in the traditional formulation of adversarial attacks, the attacker is given the chance to play second and therefore can adapt to the classifier. In other words, attacker can craft its perturbation for the particular classifier proposed by defender, while defender has to propose a classifier that should be robust to every conceivable attack. This means that if defender uses $f_1$ or $f_2$ individually, then attacker would be able to craft an attack that induces an error. In this situation, and inspired by the game of *matching pennies*, defender can turn these two strategies, that have 0 robust accuracy on their own, into a strategy that guarantees an expected accuracy of $\frac{1}{2}$ by randomizing over $f_1$ and $f_2$. Recall that attacker is optimal for any fixed classifier, **but it has no control over randomness**, which means that when faced against a

mixture of $f_1$ and $f_2$, `attacker` can compute any perturbation $\delta$ from them, but he has no knowledge of **which classifier will be used for each independent inference pass**. This can be interpreted as follows: for a given inference run, there is a chance of $\frac{1}{2}$ that the perturbation proposed by `attacker` is indeed an adversarial attack and induces an error.

Notice that if we increase the number of choices (classifiers) in matching pennies' configuration to $M > 2$, the game becomes harder for the attacker. Indeed, for every possible choice of classifier at each inference pass, there is one out of $M$ choices that lead to a successful attack (attacking the one that was sampled for the inference pass), and $M - 1$ that lead to a correct classification. An extreme example of such benefit to the defender is shown in Example 1.

## Appendix C   Details on the examples and remarks

### C.1   Example 1: Maximum mathing penny gap

Note that for this example, each classifier $w$ satisfies that $w^T x = 0$, and therefore $\mathbb{1}\{w^T x < 1\} = 1$, which means that all $w$ predict the correct label for the clean input $x$. Now we want to see that every $w$ is vulnerable at $x$.

Recall that we defined $\mathcal{H}_b$ as the space of linear classifiers $w$ such that $\|w\|_2 = \frac{1}{\epsilon}$. We can rewrite this norm as a dual norm $\|w\|_2 = \sup_{\|z\|=1} w^T z$. Moreover, this supremum is attained by some $z_w$, so for each $w$ we get $z_w$ of norm 1, such that $w^T z_w = \frac{1}{\epsilon}$.

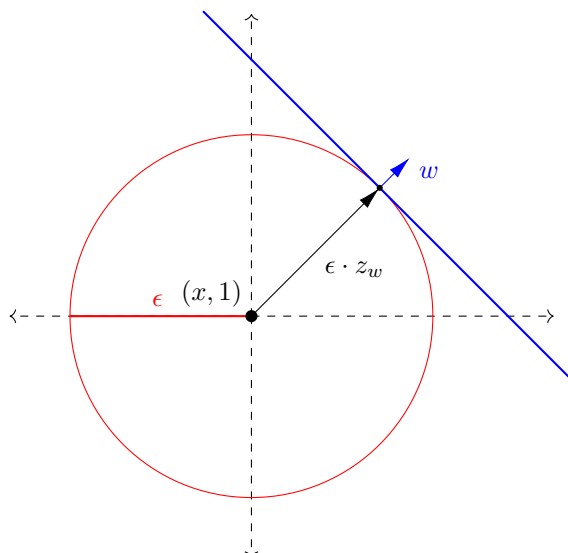

Figure 3: Visualization of Example 1. Best viewed in color. Here $w$ is one of the linear classifiers considered in $\mathcal{H}_b$. For the budget $\epsilon$, there is exactly one point $\epsilon \cdot z_w$ of intersection between the $\epsilon$-ball around $(x, 1)$ and the hyperplane defined by the equation $w^T x = 1$. All the points in the half-space $w^T x \geq 1$ are classified as 0. This includes $\epsilon \cdot z_w$, and this is the only point in the $\epsilon$-ball that belongs to this half-space. That is why $\epsilon \cdot z_w$ is the only adversarial attack that can fool $w$, and it cannot fool any other $w' \neq w$.

Now, for each $w \in \mathcal{H}_b$, consider the adversarial example $\epsilon \cdot z_w$. It is a valid adversarial example because it has norm $\epsilon$, and $w^T(\epsilon \cdot z_w) = \epsilon \cdot \frac{1}{\epsilon} = 1$, meaning that the classifier predicts $\mathbb{1}\{w^T(\epsilon \cdot z_w) < 1\} = 0$, the wrong class. The next step is to guarantee that this perturbation is unique for each $w$, which holds because we are using the Euclidean norm.

Having that $\epsilon \cdot z_w$ fools $w$ and only $w$, we can consider for simplicity $\mu$ the uniform distribution over $\mathcal{H}_b$. Let us compute the sets $\mathcal{H}_{vb}(x, 1)$ and $\mu^{\max}(x, 1) = 0$ to be able to compute the matching penny gap for this example.

As we just saw, every $w$ is itself vulnerable. This means that $\mathcal{H}_{vb}(x, 1) = \mathcal{H}_b$, and therefore $\mu(\mathcal{H}_{vb}(x, 1)) = 1$.

Given the unicity of $z_w$, we know that no two classifiers can be fooled by the same perturbation. Therefore the family of simultaneously vulnerable classifiers only contains singletons $\{w\}$. As $\mu(\{w\}) = 0$ for every $w$, taking the supremum gives $\mu^{\max}(x, 1) = 0$.

Finally, expected the matching penny gap for this example with only one point is

$$\mathop{\mathbb{E}}_{x,y}\left[\pi_{\mathbf{h}_\mu}(x, 1)\right] = \pi_{\mathbf{h}_\mu}(x, 1) = \mu(\mathcal{H}_{vb}(x, 1)) - \mu^{\max}(x, 1) = 1 - 0 = 1.$$

In other words, the mixture in this example has the best adversarial risk possible, even though it is built from classifiers with the worst adversarial risk possible individually.

## C.2 Example 2: Minimum mathing penny gap / Parallel sets

To simplify the example, consider the compact set $A = \{0\} \subset \mathbb{R}$. Then, the family of all parallel classifiers $h_r = A \oplus B_r(0)$ are the classifiers of the form $h_r(x) = \mathbb{1}\{x \in (-r, r)\}$ for $r > 0$.

Recall that a matching penny configuration arises when 1) **both classifiers are vulnerable**, but 2) **not simultaneously**. That is, each one of them must be vulnerable, but no allowed perturbation can induce an error on both at the same time. We will see that this cannot happen with this family of classifiers that are "parallel".

W.l.o.g, take any point $x > 0$ and suppose it is of class is 0. Take any two classifiers $h_{r_1}, h_{r_2}$ with $r_1 < r_2$ and fix the attacker budget to $\epsilon$. Note that $h_{r_1}$ is vulnerable at $x$ if an only if $x - \epsilon \leq r_1$. That is, the attacker must be able to move $x$ inside $(-r, r)$ with its budget of $\epsilon$. This also holds for $h_{r_2}$.

Note that to satisfy the condition that **both classifiers are vulnerable**, we must ensure that $x - \epsilon \leq r_1$ and $x - \epsilon \leq r_2$. However, any $x$ that satisfies $x - \epsilon \leq r_1$ immediately satisfies $x - \epsilon \leq r_2$, as $r_1 < r_2$, meaning that the same perturbation induces an error on both classifiers. In conclusion, they cannot be both **individually vulnerable** without being **simultaneously vulnerable** at $x$.

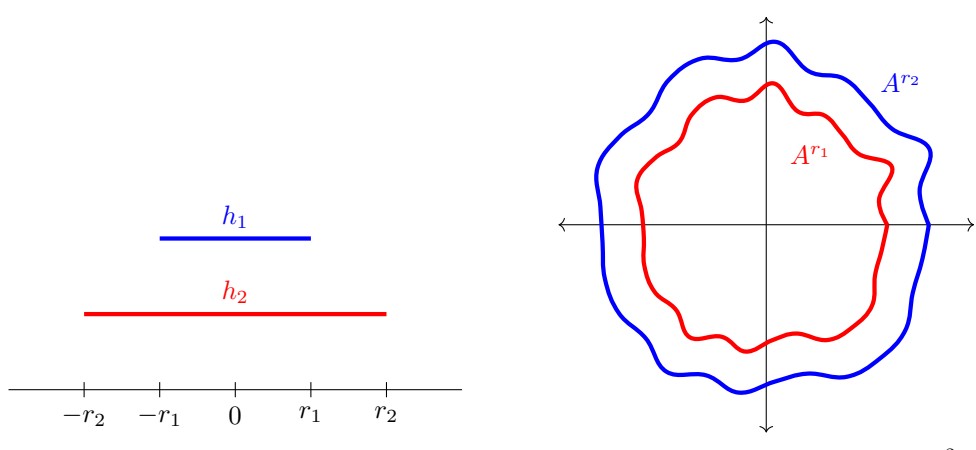

(a) Example of parallel classifiers in $\mathbb{R}$      (b) Example of parallel classifiers in $\mathbb{R}^2$

For the case of general parallel sets $A^{r_1}, A^{r_2}$, note that if $r_1 < r_2$ then $A^{r_1} \subset A^{r_2}$. Assuming that the classifiers predict 1 if the point is inside the set $A^r$, then the condition **both classifiers are vulnerable** means that a point $x$ of class 0 is at a distance of at most $\epsilon$ from both $A^{r_1}, A^{r_2}$. However, for any given $x$, being $\epsilon$-close to $A^{r_1}$ implies that $x$ is $\epsilon$-close to $A^{r_2}$, so any adversarial example of $A^{r_1}$ will be also adversarial for $A^{r_2}$. In conclusion, they can never be in a matching penny configuration.

## C.3 Multiple optimal classifiers in matching penny configuration

In this section, we are going to explain in more detail Remark 2. First, we rewrite Equation (9) which states the condition under which a mixture outperforms the individual classifiers composing it:

$$\mathop{\mathbb{E}}_{(x,y)\sim\rho}\left[\pi_{\mathbf{h}_{\mu'}}(x, y)\right] > \mathop{\mathbb{E}}_{h\sim\mu'}[\mathcal{R}_\epsilon(h)] - \inf_{h\in\mathcal{H}_b}\mathcal{R}_\epsilon(h)$$

For simplicity, assume $\mathcal{H}_b$ consist of two different classifiers $h_1, h_2$ **with the same adversarial risk** and with a **strictly positive matching penny gap**, i.e. $\mathbb{E}_{(x,y)\sim\rho}[\pi_{\mathbf{h}_\mu}(x, y)] > 0$ for some

$\mu = (\mu(h_1), \mu(h_2))$. Then the REC $\mathbf{h}_\mu$ that consists of these two classifiers will outperform each one of them. This is because the LHS of equation (9) is positive, while the RHS is exactly 0 because both classifiers have the same risk, so their average risk is equal to the best risk among them. If we add the assumption that $h_1$ and $h_2$ were optimal deterministic classifiers from a larger family $\mathcal{H}_b$, we would have created a mixture that outperforms all classifiers in it.

If now we try to soften the assumptions on $h_1, h_2$, and say that they have very similar adversarial risks, i.e. $\mathcal{R}_\epsilon(h_2) = \mathcal{R}_\epsilon(h_1) + \delta$ for some small $\delta$, then the RHS of (9) is no longer 0 but $\frac{\delta}{2}$. In this case, for condition (9) to hold we need the matching penny gap to be not only strictly positive, but greater than $\frac{\delta}{2}$.

The important intuition is that mixing classifiers that have very different adversarial risks is not ideal, as one will then need a much greater expected matching penny gap in order to actually outperform the best individual classifier. Another interesting intuition is that one can create a good mixture from very bad classifiers, as was seen in Example 1, where individual classifiers had adversarial risk of 1 (the worst possible), but mixing them resulted in an adversarial risk of 0 (the best possible). Even if Example 1 is merely a toy example, it highlights the intuition that it is possible for many non-robust classifiers to gain robustness as a mixture if they interact nicely between them and the dataset (have a high expected matching penny gap).

### C.4   Example with linear classifiers: Mixtures can improve robustness (Figure 1b)

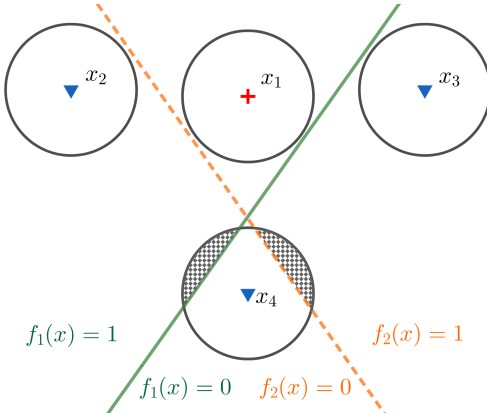

Figure 5: The mixture of $f_1$ and $f_2$ with uniform weights has lower adversarial risk than any linear classifier, motivating the use of mixtures as a robust alternative.

#### C.4.1   Detailed description of the example

Here is the exact description of the example that was introduced in Figure 1b. We will use colors to simplify the distinction between points. The points have the following coordinates: $x_1 = (0, 1), x_2 = (-2.7, 1.1), x_3 = (2.7, 1.1), x_4 = (0, -2)$. For this extended analysis, it will be easier to work with classes in $\{-1, 1\}$, so we have labels $y_1 = 1$ and $y_2 = y_3 = y_4 = -1$. We set $\epsilon = 1$.

Let $f = (w, b)$ be a generic linear classifier in $\mathbb{R}^2$, where $w = (w_1, w_2)$ is the normal vector of the hyperplane and $b$ is the bias term. The classification is done using the rule $sign(f(x)) = sign(w^T x + b)$, as in standard binary classification with labels $\{-1, 1\}$.

In this setting, a point $(x, y) \in \mathbb{R}^2 \times \{-1, 1\}$ is correctly classified if $y \cdot f(x) > 0$. Moreover, for linear classifiers, the optimal adversarial perturbation is known and robustness at a point can be easily studied. Note that given a perturbation $\delta$, we can check if it induces a misclassification by simply computing $y \cdot f(x + \delta)$. We can say $f$ is robust at $(x, y)$ if, for every perturbation of norm at most $\epsilon$,

we have that $y \cdot f(x + \delta) > 0$. This can be developed to get

$$
\begin{aligned}
y \cdot f(x + \delta) > 0 &\iff y(w^T(x + \delta) + b) > 0 \\
&\iff y(w^T x + b) + y(w^T \delta) > 0 \\
&\iff y(w^T x + b) > -y(w^T \delta)
\end{aligned}
\tag{17}
$$

For $f$ to be robust at $(x, y)$, Equation (17) must hold for all $\delta$, in particular for the quantity maximizing the RHS. This happens for $\delta^* = -y\epsilon \frac{w}{\|w\|}$. Inserting this worst case perturbation, we obtain a simple condition for robustness in the linear case:

$$
y(w^T x + b) > \epsilon \|w\|
\tag{18}
$$

Without loss of generality, we can assume $w$ has norm 1 so the last equation further simplifies. We can use these inequalities to show that no linear classifier can be robust at $x_1$, $x_2$ and $x_4$ at the same time. Indeed, being robust at each point gives us the following inequalities:

$$
\begin{aligned}
(x_1): \ & w_2 + b > 1 && \implies b > 1 - w_2 \\
(x_2): \ & 2.7w_1 - 1.1w_2 - b > 1 && \implies b < 2.7w_1 - 1.1w_2 - 1 \\
(x_4): \ & 2w_2 - b > 1 && \implies b < 2w_2 - 1
\end{aligned}
$$

Using the inequalities for $x_1$ and $x_4$, we get the condition $w_2 > \frac{2}{3}$. Using the inequalities for $x_1$ and $x_2$ gives us that $2.7w_1 - 0.1w_2 > 2$. Together with the bound on $w_2$, we get the bound $w_1 > \frac{62}{81}$. These two bounds make it impossible for $w$ to have norm 1, contradicting our hypothesis. An analogous reasoning shows that no linear classifier can be robust at $x_1$, $x_3$ and $x_4$ at the same time.

Knowing that a linear classifier can't be robust at $x_1$, $x_2$ and $x_4$ simultaneously, we can further ask what is the best adversarial risk one can get. One can easily check that it is possible to be robust on any of the pairs $(x_1, x_2)$, $(x_2, x_4)$ or $(x_1, x_2)$. At each time, robustness on the two selected points comes at the expense of non-robustness on the third point. This fact discards the strategy of being robust at the pair $(x_2, x_4)$, because being non-robust at $x_1$ implies a risk of $\frac{1}{2}$, way higher than the risk of $\frac{1}{6}$ one would have to pay for being non-robust at $x_2$ or $x_4$. With this being said, the other two solutions are optimal in terms of adversarial risk for linear classifiers, with an adversarial risk of $\frac{1}{3}$. Nevertheless, for building a robust mixture, only one of them will be useful.

### C.4.2  Matching pennies situation.

In order to show that the *matching pennies* situation can exist on $x_4$, the simplest thing is to propose two linear classifiers for which it happens. We propose the following linear classifiers:

- $f_1 = (w_1, b_1)$ with $w_1 = (0.825, 0.565132728)$ and $b_1 = 0.536876091$.
- $f_2 = (w_2, b_2)$ with $w_2 = (-0.825, 0.565132728)$ and $b_2 = 0.536876091$.

Using Equation (18) one can check that these classifiers match the situation that is illustrated in Figure 5, *i.e.* they are both robust at $x_1$, they are robust at $x_3$ and $x_2$ respectively, they are non-robust at $x_4$ but they can't be attacked on the same region, as their intersections with the $1-$ball around $x_4$ are disjoint (gray circular sectors in Figure 5). Let us denote $\mathbf{h}$ the REC $f_1$ and $f_2$ with probabilities $(\frac{1}{2}, \frac{1}{2})$.

### C.4.3  Adversarial risk calculation for the mixture of $f_1$ and $f_2$.

For $x_2$, $f_1$ is not robust, meaning its adversarial 0-1 loss on $x_2$ is 1. On the other hand, $f_2$ is robust on $x_2$, as no perturbation of norm at most $\epsilon$ will make it predict the wrong class. For these reasons, the adversarial 0-1 loss on $x_2$ for $\mathbf{h}$ is $\frac{1}{2}$. The same goes for $x_3$, and therefore they each add $\frac{1}{12}$ to the total adversarial risk (mass of each point times the 0-1 loss).

At the point $x_4$, given the *matching pennies* situation, the adversarial 0-1 for $\mathbf{h}$ is $\frac{1}{2}$, for a total adversarial risk of $\frac{1}{12}$. As both $f_1$ and $f_2$ are robust on $x_1$, the adversarial risk on $x_1$ is 0, and we can conclude that the total adversarial risk of $\mathbf{h}$ is $\frac{3}{12} = \frac{1}{4}$, which is less than the optimal adversarial risk when considering linear classifiers only, that is $\frac{1}{3}$.

## C.5   On the toy example in Trillos et al. [45, Section 5.2]

The example shown in Trillos et al. [45, Section 5.2] involves three points of different classes. We are interested in Case 4-(i), in which each pair of points can be merged into one by the adversary, but the three cannot be merged because their three $\epsilon$-balls have an empty intersection, as depicted in Figure 6. For simplicity, we assume each point has a probability $\omega_i$ of $\frac{1}{3}$, which satisfies the condition $\omega_1 < \omega_2 + \omega_3$ on Case 4 - (i) in [45].

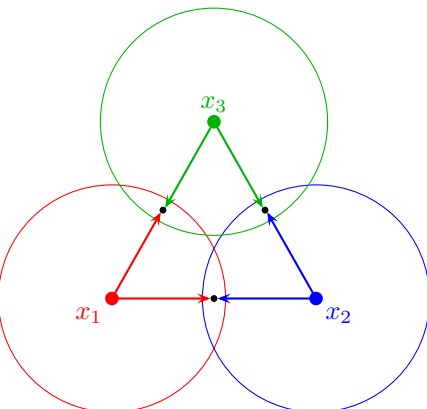

Figure 6: Toy example in Trillos et al. [45, Section 5.2], Case 4-(i). Best viewed in color.

Authors show that in this case, the optimal attack consist on distributing the mass of the original points into the two neighbors in such a way that, for each of the barycenters $\bar{x}_{ij}$, the mass coming from each of the original points $x_i$ and $x_j$ is the same. In our simplified example, this means that each original point sends $\frac{1}{6}$ of its mass to each neighboring barycenter, as depicted in Figure 7a. When it comes to an optimal classifier, the one presented in Figure 7b is optimal and coincides with the one described in the original work of [45]. This classifier predicts the original class for each $x_i$ inside the part of the ball $B_\epsilon(x_i)$ that is not intersecting any of the other balls. For each intersection $B_\epsilon(x_j) \cap B_\epsilon(x_k)$, the classifier predicts class $j$ and $k$ with probability $\frac{1}{2}$ each, and the third class $i$ with probability 0. This classifier achieves an adversarial risk of $\frac{1}{2}$, because at any point of the new adversarial distribution that is supported on $\{\bar{x}_{12}, \bar{x}_{13}, \bar{x}_{23}\}$, this classifier has probability $\frac{1}{2}$ of predicting the correct class.

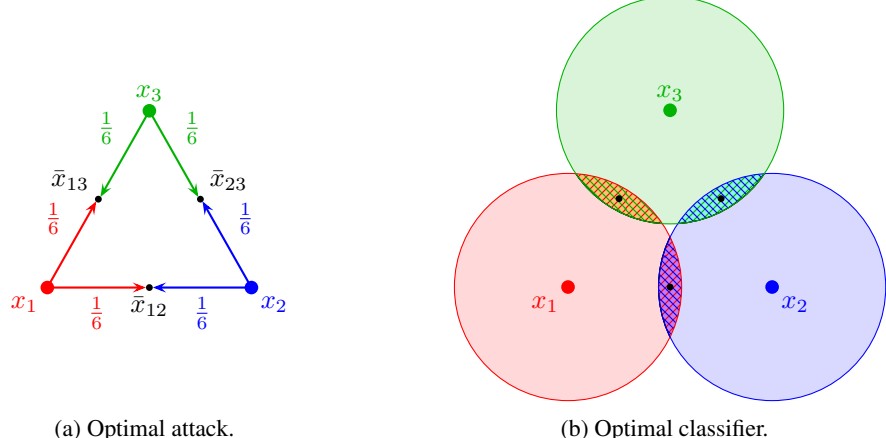

(a) Optimal attack.                                    (b) Optimal classifier.

Figure 7: Solution for toy example in Trillos et al. [45, Section 5.2], Case 4-(i). Best viewed in color.

From the perspective of our work, it is interesting that the optimal probabilistic classifier shown in Figure 7b can be built as a uniform REC of 6 deterministic classifiers $f_{ijk}$ that predict $i$ in $B_\epsilon(x_i)$, $j$ in $B_\epsilon(x_j) \setminus B_\epsilon(x_i)$ and $k$ in $B_\epsilon(x_k) \setminus (B_\epsilon(x_i) \cup B_\epsilon(x_j))$ (see Figure 8). Each $f_{ijk}$ has standard accuracy of 1, and adversarial accuracy of $\frac{1}{3}$ (risk of $\frac{2}{3}$) against an optimal attack, like the constant

classifiers, because it can only be robust at one point at most. They are optimal when restricting to deterministic classifiers. Additionally, the uniform mixture of these 6 classifiers coincides with the classifier presented in Figure 7b, which has standard accuracy of 1, and adversarial accuracy of $\frac{1}{2}$, meaning an increase in performance of $\frac{1}{2} - \frac{1}{3} = \frac{1}{6}$ with respect to the deterministic $f_{ijk}$.

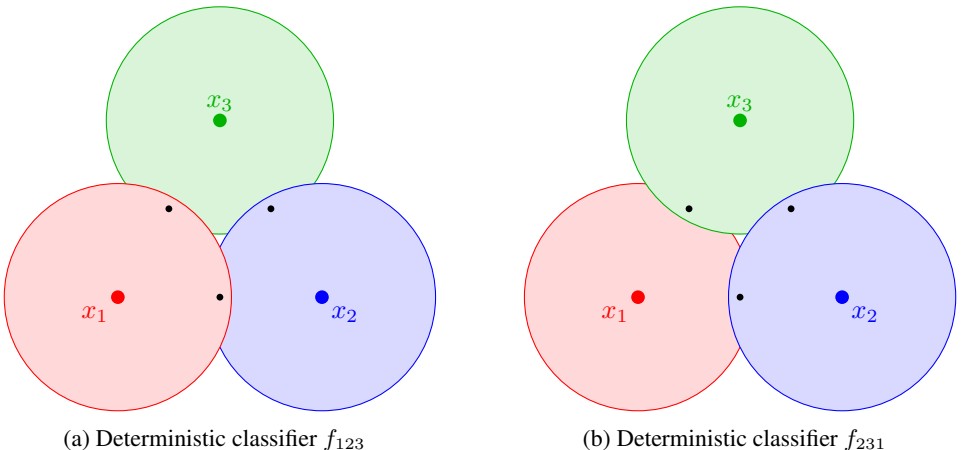

(a) Deterministic classifier $f_{123}$          (b) Deterministic classifier $f_{231}$

Figure 8: Examples of deterministic classifiers that allow the construction of the optimal classifier as a REC

One can compute the matching penny gap of this mixture of 6 classifiers in this dataset as follows: at each original point $x_i$, there are only two classifiers that are robust ($f_{ijk}$ and $f_{ikj}$), which means $\mu(\mathcal{H}_{vb}(x_i, i)) = \frac{4}{6} = \frac{2}{3}$. On the other hand, a single perturbation will not fool all the four vulnerable models. Starting from $x_i$, the attacker might move this point towards $x_j$ or $x_k$. Any of these perturbations will fool three of the six classifiers. For example, the attack $x_1 \to \bar{x}_{12}$ will fool the subset $\{f_{213}, f_{231}, f_{321}\}$, while the other three classifiers will correctly predict the class 1 at $\bar{x}_{12}$. In conclusion, $\mu^{\max}(x_i, i) = \frac{3}{6} = \frac{1}{2}$. We conclude that at each point, the matching penny gap is exactly $\frac{2}{3} - \frac{1}{2} = \frac{1}{6}$, for a total expected matching penny gap of $\frac{1}{6}$ (recall every point had the same mass). The average risk is $\frac{2}{3}$, and the risk of the mixture can be found using Equation (6) from Theorem 3.2 as $\frac{2}{3} - \frac{1}{6} = \frac{1}{2}$.

## Appendix D    Proofs

**Theorem 3.1.** For a probabilistic classifier $\mathbf{h}_\mu : \mathcal{X} \to \mathcal{P}(\mathcal{Y})$ constructed from a BHS $\mathcal{H}_b$ using any $\mu \in \mathcal{P}(\mathcal{H}_b)$, we have $\mathcal{R}_\epsilon(\mathbf{h}_\mu) \leq \mathbb{E}_{h \sim \mu}[\mathcal{R}_\epsilon(h)]$.

*Proof.* For any $\mu \in \mathcal{P}(\mathcal{H}_b)$, we have the following.

$$
\mathcal{R}_\epsilon(\mathbf{h}_\mu) = \mathbb{E}_{(x,y) \sim \rho}\left[\sup_{x' \in B_\epsilon(x)} \ell^{0\text{-}1}((x', y), \mathbf{h}_\mu)\right] = \mathbb{E}_{(x,y) \sim \rho}\left[\sup_{x' \in B_\epsilon(x)} \mathbb{E}_{h \sim \mu}\left[\mathbb{1}\{h(x') \neq y\}\right]\right]
$$

$$
\leq \mathbb{E}_{(x,y) \sim \rho}\left[\mathbb{E}_{h \sim \mu}\left[\sup_{x' \in B_\epsilon(x)} \mathbb{1}\{h(x') \neq y\}\right]\right]
$$

$$
= \mathbb{E}_{h \sim \mu}\left[\mathbb{E}_{(x,y) \sim \rho}\left[\sup_{x' \in B_\epsilon(x)} \mathbb{1}\{h(x') \neq y\}\right]\right]^2
$$

$$
= \mathbb{E}_{h \sim \mu}\left[\mathcal{R}_\epsilon(h)\right].
$$

Taking infimum with respect to $\mu$ on both sides of the above inequality, we get the following.

$$
\inf_{\mu \in \mathcal{P}(\mathcal{H}_b)} \mathcal{R}_\epsilon(\mathbf{h}_\mu) \leq \inf_{\mu \in \mathcal{P}(\mathcal{H}_b)} \mathbb{E}_{h \sim \mu}[\mathcal{R}_\epsilon(h)] \tag{19}
$$

For any $h \in \mathcal{H}_b$, we may choose the Dirac measure $\mu_h$ that assigns probability 1 to $h$ in order to obtain $\inf_{\mu \in \mathcal{P}(\mathcal{H}_b)} \mathbb{E}_{h \sim \mu} [\mathcal{R}_\epsilon(h)] \leq \mathbb{E}_{h \sim \mu_h} [\mathcal{R}_\epsilon(h)] = \mathcal{R}_\epsilon(h)$. Taking infimum over $h \in \mathcal{H}_b$, we get the following.

$$\inf_{\mu \in \mathcal{P}(\mathcal{H}_b)} \mathop{\mathbb{E}}_{h \sim \mu} [\mathcal{R}_\epsilon(h)] \leq \inf_{h \in \mathcal{H}_b} \mathcal{R}_\epsilon(h). \tag{20}$$

The remaining assertion of the theorem follows by combining (19) with (20).

$\square$

**Theorem 3.2.** For a probabilistic classifier $\mathbf{h}_\mu : \mathcal{X} \to \mathcal{P}(\mathcal{Y})$ constructed from a BHS $\mathcal{H}_b$ using any $\mu \in \mathcal{P}(\mathcal{H}_b)$,

$$\mathcal{R}_\epsilon(\mathbf{h}_\mu) = \mathop{\mathbb{E}}_{h \sim \mu} [\mathcal{R}_\epsilon(h)] - \mathop{\mathbb{E}}_{(x,y) \sim \rho} [\pi_{\mathbf{h}_\mu}(x, y)]. \tag{6}$$

*Proof.* Observe that for any $h \in \mathcal{H}_b$, $\sup_{x' \in B_\epsilon(x)} \mathbb{1}\{h(x') \neq y\} = 1$ if and only if $h \in \mathcal{H}_{vb}(x, y)$. Hence,

$$\mathop{\mathbb{E}}_{h \sim \mu} \left[ \sup_{x' \in B_\epsilon(x)} \mathbb{1}\{h(x') \neq y\} \right] = \mathop{\mathbb{E}}_{h \sim \mu} [\mathbb{1}\{h \in \mathcal{H}_{vb}(x, y)\}] = \mu(\mathcal{H}_{vb}(x, y)).$$

Taking expectation on both sides with respect to $(x, y) \sim \rho$, we get

$$\mathop{\mathbb{E}}_{(x,y) \sim \rho} [\mu(\mathcal{H}_{vb}(x, y))] = \mathop{\mathbb{E}}_{(x,y) \sim \rho} \mathop{\mathbb{E}}_{h \sim \mu} \left[ \sup_{x' \in B_\epsilon(x)} \mathbb{1}\{h(x') \neq y\} \right] = \mathop{\mathbb{E}}_{h \sim \mu} [\mathcal{R}_\epsilon(h)], \tag{21}$$

where the second equality follows from switching the order of the two preceding expectations.

For any $x' \in B_\epsilon(x)$,

$$\mathop{\mathbb{E}}_{h \sim \mu} [\mathbb{1}\{h(x') \neq y\}] = \mu(\{h \in \mathcal{H}_b : h(x') \neq y\}) \leq \mu^{\max}(x, y),$$

where the above inequality holds because the $\mu$ measure of any subset of $\mathcal{H}_b$ that is simultaneously vulnerable at some $x' \in B_\epsilon(x)$ is at most $\mu^{\max}(x, y)$. Taking supremum over all $x' \in B_\epsilon(x)$ in the above inequality, we get the following.

$$\sup_{x' \in B_\epsilon(x)} \mathop{\mathbb{E}}_{h \sim \mu} [\mathbb{1}\{h(x') \neq y\}] \leq \mu^{\max}(x, y).$$

We will now show that the above inequality also holds in the other direction.

Let $\{\mathcal{H}^k\}_{k=1}^\infty$ be a sequence of sets, $\mathcal{H}^k \in \mathfrak{H}_{svb}(x, y)$, such that $\lim_{k \to \infty} \mu(\mathcal{H}^k) = \mu^{max}(x, y)$. For each $\mathcal{H}^k$, we have by definition of $\mathfrak{H}_{svb}(x, y)$ that there exists some $x^k \in B_\epsilon(x)$ such that all classifiers $h \in \mathcal{H}^k$ are fooled by $x^k$. In other words, the total mass of classifiers that are fooled by $x^k$ is greater or equal to $\mu(\mathcal{H}^k)$. This gives

$$\sup_{x' \in B_\epsilon(x)} \mathop{\mathbb{E}}_{h \sim \mu} [\mathbb{1}\{h(x') \neq y\}] \geq \mathop{\mathbb{E}}_{h \sim \mu} [\mathbb{1}\{h(x^k) \neq y\}] \geq \mu(\mathcal{H}^k).$$

This is true for every $k$, so taking $k \to \infty$ we get that $\sup_{x' \in B_\epsilon(x)} \mathbb{E}_{h \sim \mu} [\mathbb{1}\{h(x') \neq y\}] \geq \lim_{k \to \infty} \mu(\mathcal{H}^k) = \mu^{max}(x, y)$.

Taking expectation on both sides with respect to $(x, y) \sim \rho$, we get

$$\mathcal{R}_\epsilon(\mathbf{h}_\mu) = \mathop{\mathbb{E}}_{(x,y) \sim \rho} \left[ \sup_{x' \in B_\epsilon(x)} \mathop{\mathbb{E}}_{h \sim \mu} [\mathbb{1}\{h(x') \neq y\}] \right] = \mathop{\mathbb{E}}_{(x,y) \sim \rho} [\mu^{max}(x, y)]. \tag{22}$$

Combining (22) and (21) yields (6). $\square$

---

[2]We can swap expectations by Fubini-Tonelli's theorem because the function $\sup_{x' \in B_\epsilon(x)} \mathbb{1}\{h(x') \neq y\}$ is universally measurable if $h$ is measurable (See [16, 40]) and both measure spaces $\mathcal{X} \times \mathcal{Y}$, $\mathcal{H}_b$ are assumed $\sigma$-finite. We also assume the measurability w.r.t $h$. See Appendix A for details.

**Corollary 3.1.** For $\mu' \in \mathcal{P}(\mathcal{H}_b)$, $\mathcal{R}_\epsilon(\mathbf{h}_{\mu'}) < \inf_{h \in \mathcal{H}_b} \mathcal{R}_\epsilon(h)$ if and only if the following condition holds.

$$\mathop{\mathbb{E}}_{(x,y) \sim \rho} [\pi_{\mathbf{h}_{\mu'}}(x, y)] > \mathop{\mathbb{E}}_{h \sim \mu'}[\mathcal{R}_\epsilon(h)] - \inf_{h \in \mathcal{H}_b} \mathcal{R}_\epsilon(h) \tag{9}$$

Additionally, $\inf_{\mu \in \mathcal{P}(\mathcal{H}_b)} \mathcal{R}_\epsilon(\mathbf{h}_\mu) < \inf_{h \in \mathcal{H}_b} \mathcal{R}_\epsilon(h)$ if and only if there exists $\mu' \in \mathcal{P}(\mathcal{H}_b)$ for which (9) holds.

*Proof.* Suppose (9) holds for some $\mu' \in \mathcal{P}(\mathcal{H}_b)$. Then,

$$\begin{aligned}
\inf_{\mu \in \mathcal{P}(\mathcal{H}_b)} \mathcal{R}_\epsilon(\mathbf{h}_\mu) \leq \mathcal{R}_\epsilon(\mathbf{h}_{\mu'}) &= \mathop{\mathbb{E}}_{h \sim \mu'}[\mathcal{R}_\epsilon(h)] - \mathop{\mathbb{E}}_{(x,y) \sim \rho}[\pi_{\mathbf{h}_{\mu'}}(x, y)] \\
&< \inf_{h \in \mathcal{H}_b} \mathcal{R}_\epsilon(h) + \mathop{\mathbb{E}}_{(x,y) \sim \rho}[\pi_{\mathbf{h}_{\mu'}}(x, y)] - \mathop{\mathbb{E}}_{(x,y) \sim \rho}[\pi_{\mathbf{h}_{\mu'}}(x, y)] \\
&= \inf_{h \in \mathcal{H}_b} \mathcal{R}_\epsilon(h),
\end{aligned}$$

where the first equality follows from Theorem 3.2 and the second inequality follows from the assumption in (9). Suppose (9) does not hold for $\mu' \in \mathcal{P}(\mathcal{H}_b)$. Then,

$$\begin{aligned}
\mathcal{R}_\epsilon(\mathbf{h}_{\mu'}) &= \mathop{\mathbb{E}}_{h \sim \mu'}[\mathcal{R}_\epsilon(h)] - \mathop{\mathbb{E}}_{(x,y) \sim \rho}[\pi_{\mathbf{h}'_\mu}(x, y)] \\
&\geq \inf_{h \in \mathcal{H}_b} \mathcal{R}_\epsilon(h) + \mathop{\mathbb{E}}_{(x,y) \sim \rho}[\pi_{\mathbf{h}_{\mu'}}(x, y)] - \mathop{\mathbb{E}}_{(x,y) \sim \rho}[\pi_{\mathbf{h}_{\mu'}}(x, y)] \\
&= \inf_{h \in \mathcal{H}_b} \mathcal{R}_\epsilon(h).
\end{aligned}$$

Suppose (9) does not hold for any $\mu' \in \mathcal{P}(\mathcal{H}_b)$, then taking infimum with respect to $\mu' \in \mathcal{P}(\mathcal{H}_b)$ in the above inequality, we get $\inf_{\mu' \in \mathcal{P}(\mathcal{H}_b)} \mathcal{R}_\epsilon(\mathbf{h}_{\mu'}) \geq \inf_{h \in \mathcal{H}_b} \mathcal{R}_\epsilon(h)$. $\qquad \square$

**Lemma 4.1.** Let $\mathbf{h} : \mathcal{X} \to [0, 1]$ be any measurable function. For any $\succ \in \{>, \geq\}$, the following inequality holds, and it becomes an equality if $\mathbf{h}$ is continuous or takes finitely many values:

$$\mathbb{1}\left\{ \left( \sup_{x' \in B_\epsilon(x)} \mathbf{h}(x') \right) \succ \alpha \right\} \geq \sup_{x' \in B_\epsilon(x)} \mathbb{1}\{\mathbf{h}(x') \succ \alpha\} \tag{11}$$

*Proof.* As both functions only take the values 0 and 1, it suffices to show that if the RHS is equal to 1, then so is the LHS. Suppose $\sup_{x' \in B_\epsilon(x)} \mathbb{1}\{\mathbf{h}(x') \succ \alpha\} = 1$. As the function $\mathbb{1}\{\mathbf{h}(x') \succ \alpha\}$ takes only a finite number of values, this implies that there exists some $x^* \in B_\epsilon(x)$ such that $\mathbb{1}\{\mathbf{h}(x*) \succ \alpha\} = 1$. This means that $\mathbf{h}(x*) \succ \alpha$, and therefore $\sup_{x' \in B_\epsilon(x)} \mathbf{h}(x') \succ \alpha$. This makes the LHS equal to 1.

If we further assume that $\mathbf{h}$ is continuous or takes a finite number of values, then if the LHS is equal to one, $\qquad \square$

**Theorem 4.1.** Let $\mathbf{h} : \mathcal{X} \to [0, 1]$ be any probabilistic binary classifier. Let $h^\alpha$ be the $\alpha$-*threshold* classifier. Then the following equation holds:

$$\mathcal{R}_\epsilon(\mathbf{h}) \geq \int_0^1 \mathcal{R}_\epsilon(h^\alpha) d\alpha \geq \inf_\alpha \mathcal{R}_\epsilon(h^\alpha). \tag{12}$$

Further, if $\mathbf{h}$ is either continuous or takes finitely many values, the first inequality in (12) becomes an equality.

*Proof.* We begin the proof by rewriting the adversarial risk of $\mathbf{h}$ in terms of the adversarial risk for each class. To alleviate notation, we denote $\ell_\epsilon^{0\text{-}1}((x, y), \mathbf{h}) = \sup_{x' \in B_\epsilon(x)} \ell^{0\text{-}1}((x', y), \mathbf{h})$. Recall from Equation (1) that

$$\mathcal{R}_\epsilon(\mathbf{h}) \quad = \nu(0) \cdot \mathcal{R}_\epsilon^0(\mathbf{h}) + \nu(1) \cdot \mathcal{R}_\epsilon^1(\mathbf{h}) \tag{23}$$

Let us now develop the terms $R_\epsilon^y(\mathbf{h})$. For any $\succ \in \{>, \geq\}$ we obtain the following:

$$R_\epsilon^y(\mathbf{h}) = \mathop{\mathbb{E}}_{x \sim p_y} \left[ \ell_\epsilon^{0\text{-}1}((x,y), \mathbf{h}) \right]$$

$$= \mathop{\mathbb{E}}_{x \sim p_y} \left[ \int_0^1 \mathbb{1}\{\ell_\epsilon^{0\text{-}1}((x,y), \mathbf{h}) \succ \alpha\} d\alpha \right] \tag{24}$$

$$= \int_0^1 \mathop{\mathbb{E}}_{x \sim p_y} \left[ \mathbb{1}\{\ell_\epsilon^{0\text{-}1}((x,y), \mathbf{h}) \succ \alpha\} \right] d\alpha$$

In the last equation, we were able to interchange the two integrals using Tonnelli's theorem. Indeed, the function $(x,\alpha) \mapsto \mathbb{1}\{\ell_\epsilon^{0\text{-}1}((x,y), \mathbf{h}) \succ \alpha\}$ is Lebesgue measurable in $\mathcal{X} \times \mathbb{R}$ if $\ell_\epsilon^{0\text{-}1}((\cdot, 0), \mathbf{h})$ is Lebesgue measurable, which is the case [40, Theorem 1].

The next step is to use Lemma 4.1 to interchange the supremum operator and the indicator function. This will make the adversarial risk of the $h^\alpha$ appear. For $y = 0$ and replacing the operator $\succ$ by $>$, we obtain the following:

$$R_\epsilon^0(\mathbf{h}) = \int_0^1 \mathop{\mathbb{E}}_{x \sim p_0} \left[ \mathbb{1}\{\ell_\epsilon^{0\text{-}1}((x,0), \mathbf{h}) > \alpha\} \right] d\alpha$$

$$\geq \int_0^1 \mathop{\mathbb{E}}_{x \sim p_0} \left[ \sup_{x' \in B_\epsilon(x)} \mathbb{1}\{\mathbf{h}(x') > \alpha\} \right] d\alpha \tag{25}$$

$$\geq \int_0^1 \mathcal{R}_\epsilon^0(h^\alpha) d\alpha$$

Now we do a similar development for $y = 1$ and replacing the operator $\succ$ by $\geq$ to obtain:

$$R_\epsilon^1(\mathbf{h}) \geq \int_0^1 \mathop{\mathbb{E}}_{x \sim p_1} \left[ \sup_{x' \in B_\epsilon(x)} \mathbb{1}\{1 - \mathbf{h}(x') \geq \alpha\} \right] d\alpha$$

$$\geq \int_0^1 \mathop{\mathbb{E}}_{x \sim p_1} \left[ \sup_{x' \in B_\epsilon(x)} \mathbb{1}\{1 - \alpha \geq \mathbf{h}(x')\} \right] d\alpha$$

$$\geq \int_0^1 \mathop{\mathbb{E}}_{x \sim p_1} \left[ \sup_{x' \in B_\epsilon(x)} 1 - \mathbb{1}\{\mathbf{h}(x') > 1 - \alpha\} \right] d\alpha \tag{26}$$

$$\geq \int_0^1 \mathop{\mathbb{E}}_{x \sim p_1} \left[ \sup_{x' \in B_\epsilon(x)} 1 - \mathbf{h}^{1-\alpha}(x') \right] d\alpha$$

$$\geq \int_0^1 \mathop{\mathbb{E}}_{x \sim p_1} \left[ \sup_{x' \in B_\epsilon(x)} \ell^{0\text{-}1}((x,1), \mathbf{h}^{1-\alpha}) \right] d\alpha$$

$$\geq \int_0^1 \mathcal{R}_\epsilon^1(\mathbf{h}^{1-\alpha}) d\alpha = \int_0^1 \mathcal{R}_\epsilon^1(\mathbf{h}^u) du$$

In the last equation, the change of variable $u = 1 - \alpha$ allows us to complete the calculations and obtain the desired result. Putting everything together, we obtain the following result about the original probabilistic classifier $\mathbf{h}$:

$$\mathcal{R}_\epsilon(\mathbf{h}) \geq \int_0^1 \nu(0) \mathcal{R}_\epsilon^0(h^\alpha) + \nu(1) \mathcal{R}_\epsilon^1(h^\alpha) d\alpha = \int_0^1 \mathcal{R}_\epsilon(h^\alpha) d\alpha \tag{27}$$

In particular, Equation (27) implies that $\mathcal{R}_\epsilon(\mathbf{h}) \geq \min_\alpha \mathcal{R}_\epsilon(h^\alpha)$, meaning that for any $\mathbf{h}$ probabilistic binary classifier, there is always a deterministic $\alpha$-threshold classifier with better or equal adversarial risk. $\qquad\square$

**Corollary 5.1.** Let $\mathcal{H}_b$ be any family of deterministic binary classifiers. Let $\mathcal{M} = \mathcal{P}_M(\mathcal{H}_b) \subset \mathcal{P}(\mathcal{H}_b)$ be the subset of probability measures over $\mathcal{H}_b$ defining RECs as in Section 2.3. Let $\mathcal{A} = \{h^{-1}(1) : h \in \mathcal{H}_b\}$. If $\mathcal{A}$ is closed under union and intersection, then

$$\inf_{h \in \mathcal{H}_b} \mathcal{R}_\epsilon(h) = \inf_{\mu \in \mathcal{P}_M(\mathcal{H}_b)} \mathcal{R}_\epsilon(\mathbf{h}_\mu).$$

*Proof.* Theorem 4.1 applied to RECs (see Section 4.2) tells us that $\overline{\mathcal{H}_b}(\mathcal{P}_M(\mathcal{H}_b))$ is the set of all weighted ensembles built from $\mathcal{H}_b$. It is clear that for any $h \in \mathcal{H}_b$, $\delta_h \in \mathcal{P}_M(\mathcal{H}_b)$, so $\mathcal{H}_b \subseteq \overline{\mathcal{H}_b}(\mathcal{P}_M(\mathcal{H}_b))$. Let us now show the other inclusion.

Take any $\mu \in \mathcal{P}_M(\mathcal{H}_b)$ and consider any weighted ensemble $h^\alpha$ over $\mathcal{H}_b$ of the form $h^\alpha(x) = \mathbb{1}\{\sum_{m \in [M]} p_m h_m > \alpha\}$ with $p_m = \mu(h_m)$. Define the function $g^\alpha : \{0,1\}^M \to \{0,1\}$ as $g^\alpha(z_1 \ldots z_M) = \mathbb{1}\left\{\sum_{m=1}^M p_m z_m > \alpha\right\}$. Then, $h^\alpha$ can be written as $g^\alpha(h_1(x) \ldots h_M(x))$. Because the $p_m$ are positive, the function $g^\alpha$ is a monotone boolean function. As any monotone boolean function, $g^\alpha$ can be written as a disjunctive normal form (DNF) without negations [9]. Thus, the set $A_{\text{ENS}} = \left\{x \in \mathcal{X} : h^\alpha(x) = 1\right\}$ is a union of intersections over the sets $A_1 \ldots A_M$ where $A_k = h_m^{-1}(1)$. Because $\mathcal{A}$ is closed under union and intersection, $A_{\text{ENS}} \in \mathcal{A}$, which means that $h^\alpha \in \mathcal{H}_b$. Thus, $\overline{\mathcal{H}_b}(\mathcal{P}_M(\mathcal{H}_b)) \subseteq \mathcal{H}_b$.

As all the hypothesis of Theorem 5.1 are met, we can conclude that $\inf_{h \in \mathcal{H}_b} \mathcal{R}_\epsilon(h) = \inf_{\mu \in \mathcal{P}_M(\mathcal{H}_b)} \mathcal{R}_\epsilon(\mathbf{h}_\mu)$.

$\square$

