# OpenReview forum: "On the Role of Randomization in Adversarially Robust Classification"
_NeurIPS.cc/2023/Conference — NeurIPS 2023 spotlight_

### Official Review · Reviewer_ZKUe · 2023-06-20

**Soundness:** 3 good
**Presentation:** 3 good
**Contribution:** 3 good
**Rating:** 7
**Confidence:** 3

**Summary:**

The paper investigates randomized classifiers and their robustness against adversarial examples. The authors' contributions can be found along three different axes. First, the authors give the necessary and sufficient conditions so that from a set of deterministic classifiers one can build a randomized classifier that outperforms the best deterministic classifier from the original set (w.r.t. adversarial risk). In sequence the authors present some results in the opposite direction in the binary classification case; e.g., for every randomized ensemble classifier there exists a deterministic weighted ensemble with better adversarial risk. Finally, the authors identify the conditions under which randomization can provably help on a deterministic base hypothesis set.

After rebuttal: Upgraded my score from "Weak Accept" to "Accept".

**Strengths:**

Some of the ideas of the paper are building on ideas and findings of prior work by Dbouk and Shanbhag; namely, the observation that a randomized classifier can achieve a decisive advantage over any deterministic classifier, because the adversary can only force a misclassification on a subset of all vulnerable classifiers, where "vulnerable" means classifiers that can be fooled with perturbations of certain magnitude. This observation and the continuation of the work is interesting and the paper is well-written for the largest part, though, near the end I found it more dense than what I would have preferred it to be. I expect this paper to influence to some extent the work that will be done on the robustness of randomized classifiers in the near future. Overall an interesting paper with interesting findings, though, for completeness, I would like to see the authors add some information in the supplementary material even for simple claims; e.g., clearly write down the calculations for Figure 1b, or clarify other small remarks that they have here and there. I also like Section 6; the conclusion that the authors have written -- a clear summary of their contributions and a summary of ideas related to their work that can be explored as future work.

**Weaknesses:**

Figure 1 appears 2 pages after it is first mentioned. One page difference is ok, but two pages is strange.

The authors assume that the label y does not change within the perturbation budget $\epsilon$ around $x$. While this is a reasonable assumption, and in fact it is the prevailing assumption in adversarial robustness, nevertheless, it is not the only assumption one can make. Clarifying this and citing any of the following two papers (or both?) so that someone can find more information about the topic, I believe can be important and potentially also motivate further avenues for randomized classification. The two papers are:

[M1] Adversarial Risk and Robustness: General Definitions and Implications for the Uniform Distribution
https://proceedings.neurips.cc/paper_files/paper/2018/file/3483e5ec0489e5c394b028ec4e81f3e1-Paper.pdf

[M2] On the Hardness of Robust Classification
https://www.jmlr.org/papers/volume22/20-285/20-285.pdf

Line 102: Write down explicitly the adversarial risk of a deterministic classifier in the appendix, if it does not fit in the main part of the paper, so that there is no ambiguity.

I think the "Matching Penny" notions are not clearly defined in the paper. It is apparent that some of the ideas have appeared in prior work but the authors spend nearly no time explaining the original ideas and the original context.  I believe this should change.  In particular, the authors cite two books in lines 168-170 without saying much more; the reader may have no idea about these books and this line of work.

It is quite unfortunate that Definition 1 splits between pages because the notation is a bit overloaded and it is clarified on the following page.


**Questions:**

Q1. In lines 144-146 it is stated "Consequently, we get that there always exists a probabilistic classifier that is at least as good as the best deterministic classifier in the BHS $\mathcal{H}_b$." Can you please explain why and how this follows from the previous sentence?

Q2. In lines 210-211 we have "Observe that every classifier in $H_b$ is vulnerable at $(x,y) = (0,1)$ and so $R_{\epsilon}(h) = 1$ for all $h\in H_b$". I am not sure I follow; since $x = 0$, then $w^T x = 0$ and therefore 1{$w^Tx < 1$} should return 1, which is really the label $y$. Can you explain?

**Limitations:**

I think the paper is ok in this regard, especially in the light of the clear ideas for future work that one can pursue and are listed in Section 6.

---

> ### Author Rebuttal · Authors · 2023-08-09
>
> Thank you for your thoughtful review and pointing out relevant references. We are very pleased to know you found our work "interesting and well-written for the largest part". We also share the expectation that our contribution will "influence to some extent the work that will be done on the robustness of randomized classifiers in the near future". Below we answer your questions and concerns.
>
> > The authors assume that the label $y$ does not change within the perturbation budget $\epsilon$ around $x$
>
> Yes, we indeed work on the *corrupted-instance* setting according to [1] (*constant-in-the-ball* [2]), as opposed to the *error region* or *prediction change* formulations [1]. As discussed in [1, 2], this definition can be problematic because even the true concept class can have positive and even constant adversarial risk. However, we consider the *corrupted-instance* setting because it is commonly studied in adversarial examples literature [3, 4, 5] and in the work that is closely related to ours, like [6, 7].
>
> We will include in the paper a comment mentioning the different notions of adversarial risk and clearly clarify that our work corresponds to the *corrupted-instance* setting.
>
> > I think the "Matching Penny" notions are not clearly defined in the paper
>
> Please refer to the Global response.
>
> > In lines 144-146 it is stated "Consequently, we get that there always exists a probabilistic classifier that is at least as good as the best deterministic classifier in the BHS $\mathcal{H}_b$." Can you please explain why and how this follows from the previous sentence?
>
> We thank you for pointing out this sentence, as we agree it needs to be rephrased to really transmit the correct message. There is a simpler argument to support this remark, which is expressed in the proof of Theorem 3.1 in the Supplementary Material, lines 482-486. The idea is that the original deterministic classifiers $h \in \mathcal{H}\_b$ can be seen as probabilistic classifiers by considering $\mathbf{h}\_{\mu}$ with $\mu = \delta\_{h}$, the Dirac measure over $h$. This being said, one can take infimum over all probabilistic classifiers over $\mathcal{H}_b$, and obtain that for any $h' \in \mathcal{H}_b$ the following holds:
>
> $$\inf\_{\mu \in \mathcal{P}(\mathcal{H}_b)} \mathcal{R}\_{\epsilon}(\mathbf{h}\_{\mu}) \le \mathcal{R}\_{\epsilon}(\mathbf{h}\_{\delta\_{h'}}) = \mathcal{R}\_{\epsilon}(h')$$
>
>
> > In lines 210-211 ... [About Example 1]
>
> As you state, each classifier $w$ satisfies that $w^Tx = 0$, and therefore $\mathcal{1}\\{w^Tx < 1\\} = 1$, which means that all $w$ predict the correct label for the *clean* input $x$. Now we want to see that every $w$ is vulnerable at $x$.
>
> Recall that we defined $\mathcal{H}\_b$ as the space of linear classifiers $w$ such that $\left\lVert w \right\rVert\_{\star} = \frac{1}{\epsilon}$. In our case, $\mathcal{X} = \mathbb{R}^d$ with (usually) an $L_p$ norm $\left\lVert \cdot \right\rVert$, so we can rewrite the dual norm as $\left\lVert w \right\rVert_{\star} = \sup\_{\left\lVert z \right\rVert=1} \{ w^Tz \}$. Moreover, this supremum is attained by some $z_w$, so for each $w$ we get $z_w$ of norm 1, such that $w^T z_w = \frac{1}{\epsilon}$.
>
> Now, for each $w \in \mathcal{H}_b$, consider the adversarial example $\epsilon \cdot z_w$. It is a valid adversarial example because it has norm $\epsilon$, and $w^T (\epsilon \cdot z_w )= \epsilon \cdot \frac{1}{\epsilon} = 1$, meaning that the classifier predicts $\mathcal{1}\\{w^T (\epsilon \cdot z_w) < 1\\} = 0$, the wrong class. The next step is to guarantee that this perturbation is unique for each $w$, and your question made us realize that we need to add an extra assumption for this: we need to work with the $L_2$ norm so that $z_w$ is the only vector attaining the supremum on the dual norm.
>
> Having that $\epsilon \cdot z_w$ fools $w$ and *only* $w$, we can consider for simplicity $\mu$ the uniform distribution over $\mathcal{H}_b$. Let us compute the sets $\mathcal{H}\_{vb}$ and $\mathcal{H}^{max}\_{svb}$ to be able to compute the *matching penny gap* for this example.
>
> As we just saw, every $w$ is itself vulnerable. This means that $\mathcal{H}\_{vb} = \mathcal{H}\_{b}$, and therefore $\mu(\mathcal{H}\_{vb}) = 1$.
>
> Given the unicity of $z_w$, we know that no two classifiers can be fooled by the same perturbation. Therefore the family of simultaneously vulnerable classifiers only contains singletons $\{w\}$. As $\mu(\{w\}) = 0$ for every $w$, taking the supremum gives $\mu(\mathcal{H}^{max}_{svb}) = 0$.
>
> Finally, the matching penny gap is $\pi_{\mathbf{h}\_{\mu}}(x,y) = \mu(\mathcal{H}\_{vb}) - \mu(\mathcal{H}^{max}\_{svb}) = 1$. In other words, the mixture in this example has the best adversarial risk possible, even though it is built from classifiers with the worst adversarial risk possible individually.
>
> We will add the complete explanation on the Supplementary material, and add to the main paper that the example works for the $L_2$ norm.
>
> [1] Adversarial Risk and Robustness: General Definitions and Implications for the Uniform Distribution
>
> [2] On the Hardness of Robust Classification
>
> [3] Towards Deep Learning Models Resistant to Adversarial Attacks
>
> [4] The Limitations of Deep Learning in Adversarial Settings
>
> [5] The Many Faces of Adversarial Risk
>
> [6] Adversarial Vulnerability of Randomized Ensembles
>
> [7] Mixed Nash Equilibria in the Adversarial Examples Game

---

> > ### Comment · Reviewer_ZKUe · 2023-08-15
> > **Thank you for the responses**
> >
> > I have read the other reviews as well as the response that the authors have provided to the various issues that were raised in the reviews and I am happy with the explanations.  I can see that the paper will have more clarity and better content overall in the end and I will upgrade my vote from "Weak Accept" to "Accept".
> >
> > Thank you for a very interesting paper!

---

> > > ### Author Response · Authors · 2023-08-16
> > > **Thank you**
> > >
> > > We are very glad that our response has made the paper clearer. We thank you for your upgrade !

---

### Official Review · Reviewer_Tq42 · 2023-07-03

**Soundness:** 3 good
**Presentation:** 2 fair
**Contribution:** 4 excellent
**Rating:** 7
**Confidence:** 4

**Summary:**

Several proposed adversarial defenses involve randomized classifiers. Prior theoretical work analyzes both the randomized and non-randomized cases. This paper answers: 1) When does randomization 'help' in adversarial classification? 2) When does it suffice to consider deterministic classifiers?

The authors start by describing three common paradigms ( randomized ensemble classifier (REC), weight-noise injection classifier (WNC), input-noise injection classifier (INC)) for constructing randomized classifiers. Subsequently, the authors give a condition describing when randomized classifier over a hypothesis set $\mathcal H_b$ will outperform deterministic classifiers. Next, the authors show that one can always threshhold a binary randomized classifier to get a better performing deterministic classifier. Lastly, the authors show that if $\cH_b$ is all measurable sets, then randomization over $\cH_b$ does not improve the adversarial risk.

**Strengths:**

- The paper answers and important question in the field of adversarial learning: when does randomization help with adversarial classification?
- Proofs seem simple and clear

**Weaknesses:**

The exposition is not great. Here are some specific issues.
0. Please state your threat model. Your adversary has access to the randomized classifier, but does not know which random function the learner uses every time a point is classified.
1. Equation 1:  $\mathcal R_\epsilon ^y$ is not defined before equation (1)
Note that you also need to explain that $\sup_{x'\in B_\e(x)} \ell^{0-1}((x',y),h)$ is measureable. See appendix A of the following reference for this result:
Existence and minimax theorems for adversarial surrogate risks in binary classification. N. S. Frank, J. Niles-Weed.
2. Foot note 1: I understand how such a proof would go, but I would expect to see a proof with a maximizing sequence rather than a maximizer in the supplementary material
3. I couldn't understand the argument in line 191-192
4. lines 203-204: I couldn't understand the sentence. "More generally, assumption..."
5. I had a hard time understanding the parallel decision boundaries discussion in example 2
6. line 276: I think an REC built from 2 base classifiers takes on continuously many values. Can you better explain 276-278?
7. The supplementary material is quite disorganized

**Questions:**

- line 108: The Borel sigma algebra is defined in terms of the open sets on a topological space. How are you defining the topology on the set $\mathcal H_b$? It seems that in the REC, ENC, INC paradigms, there is a "continuous" map from $\mathbb R^d$ to $\mathcal H_b$. Perhaps you could use this map.
- Can you explain the name "penny matching gap"?
- In section 4.1, I would expect that one can always take the threshhold $\alpha=1/2$. Why is this not the case?


**Limitations:**

Some of the results only hold in the binary case. The paper discusses this limitation

---

> ### Author Rebuttal · Authors · 2023-08-09
>
> Thank you for all the effort you put in reviewing our work and pointing out relevant references. We appreciate that you find that our paper "answers an important question in the field of adversarial learning". We will put all out effort in clarifying the exposition of the paper to make it clearer and avoid confusion. Below are some comments addressing your suggestions and questions. Also, as your remarks and questions were very important and numerous, we will be answering some of them in an official comment to this rebuttal if we are allowed.
>
> > Please state your threat model
>
> We will add in Section 2 the details about our threat model, in which, as you state, the adversary has full access to the models (parameters, gradients, number of models, etc.), but no control nor access over the randomness used by the defender to choose which model to use for each inference pass. That is, we are in a white-box threat model without access nor control over the random number generators.
>
> > line 108: About the Borel sigma algebra over $\mathcal{H}_b$
>
> In all our particular use cases, the space $\mathcal{H}_b$ is either finite or identifiable to some usual subset of $\mathbb{R}^d$. In the case of INC, $\mathcal{H}_b$ is identifiable with the input space $\mathcal{X} \subset \mathbb{R}^d$. For WNC, when classifiers are parametrized over some $\mathbb{R}^p$ (neural networks, linear classifiers), this is also the case. So in any of these cases, the topology considered is the usual topology and the Borel $\sigma$-algebra is defined accordingly.
>
> In Remark 4 we consider a setting closer to [1, 2, 3], where $\mathcal{H}_b$ is much more complex. However, this remark holds when one takes any finite number of optimal classifiers, so even if the family $\mathcal{H}^*$ of optimal classifiers is complicated, we can simply take any pair of them ($\mathcal{H}_b$ finite, with two elements) and say that their matching penny gap must be zero.
>
> We thank you for pointing out the need for clarification. We will make it clear in Section 2 that, for this work, $\mathcal{H}_b$ will always be either finite or directly identifiable with some subset of $\mathbb{R}^p$. Furthermore, we will also reformulate Remark 4 accordingly.
>
> > Note that you also need to explain that $\sup_{x' \in B_{\epsilon}(x) } \ell^{0 -1}((x', y), h)$ is measureable
>
> We totally agree that the measurability of the 0-1 loss under attack is non-trivial, and it is important to discuss it. Your remark sparked an important discussion within the team that led to the following conclusions:
>
> To ensure measurability, we add the assumption that for every $y$, the evaluation function $f : \mathcal{X} \times \mathcal{H}_b \to \mathbb{R}$ defined as $f(x, h) = h(x)_y$ (the $y$-th component of $h(x)$ seen as a vector function) is Borel measurable in $\mathcal{X} \times \mathcal{H}_b$. This might not be true in general (see this thread https://mathoverflow.net/a/28114), but it holds in our setting. What we mean is that in practice, $f$ takes a simpler form, like $f(x, w) = (w^T x)\_y$ in the case of linear classifiers, or $f\_{w}(x)\_y$ in the case of parametrized neural networks. In all these settings, the function $f$ can be assumed to be well-behaved function from $\mathbb{R}^d \\times \mathbb{R}^p$ to $\\mathbb{R}^K$ that satisfies the measurability condition.
>
> Assuming that for every $y$, the mapping $(x, h) \to h(x)\_y$ is Borel measurable, we can then write the expected 0-1 loss as
>
> $$\\ell^{0 -1}((\\cdot, y),\\mathbf{h}\_{\\mu}) = \\mathbb{E}\_{h \\sim \\mu} \\left[ \\mathcal{1}\\{h(x) \\ne y \\} \\right] = \\mathbb{E}\_{h \\sim \\mu} \\left[ 1 -h(x)\_y \\right] = \\int_{\\mathcal{H}\_b} 1 -h(x)\_y d\\mu(h)$$
>
> and by Fubini-Tonnelli's Theorem, this function will be Borel measurable in $\mathcal{X}$.
>
> The final step would be to apply [1, Appendix A, Theorem 27] to prove that the loss under attack is universally measurable, and that the adversarial risk is well-defined in the universal $\sigma$-algebra.
>
> > Proof with a maximizing sequence
>
> We will add this proof in the supplementary material. We will also modify the definitions in page 5 accordingly. In particular, we will define $\\mu^{max}(x, y) = \\sup\_{\\mathcal{H}' \\in \\mathcal{H}\_{svb}} \\mu(\\mathcal{H}')$ instead of using the $argmax$, which somehow assumes the existence of such $\mathcal{H}^{max}\_{svb}$ (and $x^{max}$ in the proof).
>
> > I couldn't understand the argument in line 191-192
>
> The main intuition is that the optimal attack for a mixture of models, i.e. in which the defender randomly samples a model at inference, consists exactly on crafting an attack that simultaneously attacks as many classifiers as possible. This lines however will be modified according to the last answer.
>
> > Can you explain the name "penny matching gap"?
>
> Please refer to the global response.
>
> [1] Existence and minimax theorems for adversarial surrogate risks in binary classification
>
> [2] The Many Faces of Adversarial Risk
>
> [3] On the Existence of the Adversarial Bayes Classifier (Extended Version)
>
> [4] Randomization matters. How to defend against strong adversarial attacks
>
> [5] Adversarial Risk via Optimal Transport and Optimal Couplings

---

> > ### Comment · Reviewer_Tq42 · 2023-08-13
> >
> > **measurability issue**
> > That's interesting about measurability! This heart of the issue seems to be showing that a projection of a Borel measurable function is measurable.
> >
> > Under mild assumptions, The projection of a Borel measurable function is measurable with respect to the analytic sigma algebra (which is contained in the universal sigma algebra).
> >
> > The reference [1] includes a variant of this argument for $\mathbb R^d\times S$, where $S\subset \mathbb R^d$.
> >
> > You might also be able to find the fact you need in [6]
> >
> >
> > **further references**
> > Relating to theorem 4.1: Section 5.2 of [7] provides an example of a distribution with 3 classes for which randomized classifiers preform strictly better than deterministic classifiers.
> >
> > [6] Stochastic optimal control: the discrete time case
> >
> >
> >
> > [7] The multimarginal optimal transport formulation of adverarial multiclass classification

---

> > > ### Author Response · Authors · 2023-08-14
> > > **Thanks again for the useful references**
> > >
> > > We would like to thank you for pointing out these useful references. After reading [6], we found that another way of ensuring the Borel-measurability of the projections would be to satisfy [6, Proposition 7.14]. We would need $\mathcal{X}$ to be a Borel space and $\mathcal{Y}$ a product of Borel spaces (already satisfied), and this would yield that Borel-measurability of $h: \mathcal{X} \to \mathcal{Y} \subset \mathbb{R}^K$ is enough to ensure measurability of individual components $h_y$. These assumptions are simpler and close to the settings that are used in practice.
> > >
> > > With respect to the Example in [7, Section 5.2], we find it fascinating and closely related to our setting. Thank you for pointing it out. In their case 4-i authors show that a randomized classifier that assigns $\frac{1}{2}$ to every "conflicting" point or region is a saddle point. This is closely related to the Nash equilibrium in the original game of matching pennies. Indeed, if one restricts to a pair of points, there is a matching pennies game going on: authors say "The adversary gathers classes as much as possible and distributes them as uniform as possible". That is, at each point $\bar{x}_{ij}$ there is the same probability that this point came from class $i$ and $j$, then the equilibrium strategy is to predict uniformly at random between class $i$ and $j$.
> > >
> > > It is also interesting that the optimal randomized classifier in this example (weak partition, page 39 of [7], bottom) can be built as a uniform mixture of 6 deterministic ones $f_{ijk}$ that predict $i$ in $B\_{\epsilon}(x_i)$, $j$ in $B\_{\epsilon}(x_j) \setminus B\_{\epsilon}(x_i)$ and $k$ in $B\_{\epsilon}(x_k) \setminus (B\_{\epsilon}(x_i) \cup B\_{\epsilon}(x_j))$. Each $f\_{ijk}$ has standard power of 1, and adversarial power of $w_i$ against an optimal attack, like the constant classifiers. However, the uniform mixture of these 6 classifiers would have standard power of 1, and adversarial power of 0.5, meaning an increase in performance of $0.5 - w_1 \gt 0$. One can compute the matching penny gap of this mixture of 6 classifiers in this dataset, and conclude it is $\frac{1}{6}$. The average risk is $\frac{2}{3}$, and the risk of the mixture is $\frac{2}{3} - \frac{1}{6} = \frac{1}{2}$.
> > >
> > > Lastly, this example shows that Theorem 4.1 does not hold in multiclass! Indeed, authors show that their randomized classifier is an equilibrium, and that "In fact, it is unavoidable to introduce weak partitions". They further say that "In other words, even this simple discrete measures, it is necessary to extend strong partition to weak partition in order to achieve the minimax value". We will definitely add this important remark in the paper. The conclusion would therefore be that in the multiclass setting, there exist data distributions for which randomization is necessary, proving an even stronger point for randomization for adversarial robustness. Understanding these situations would be an interest avenue of future work, because as the authors say, it "depends on both the geometry of data distributions and their magnitudes" and even with discrete toy examples, there are non-trivial situations.
> > >
> > > **Answers to other questions**
> > >
> > > > Parallel decision boundaries
> > >
> > > To simplify the example, consider $A = \\{0\\} \subset \mathbb{R}$ and the parallel classifiers $h_r = A \oplus B_r(0)$ of the form $h_r(x) = \mathcal{1}\\{ x \in (-r, r) \\}$ for $r > 0$..
> > >
> > > Recall that a matching penny configuration for two classifiers arises when 1) **both are vulnerable**, but 2) **not simultaneously**. We will see that this cannot arrive with this family of classifiers that are "parallel".
> > >
> > > W.l.o.g, take any point $x > 0$ and suppose it is of class is 0. Take any two classifiers $h\_{r_1}, h\_{r_2}$ with $r_1 < r_2$ and fix the attacker budget to $\epsilon$. Note that $h_{r_1}$ is vulnerable at $x$ if an only if $x - \epsilon \le r_1$. That is, the attacker must be able to move $x$ inside $(-r_1, r_1)$ with its budget of $\epsilon$. This also holds for $h_{r_2}$.
> > >
> > > To satisfy condition 1), we must ensure that $x - \epsilon \le r_1$ and $x - \epsilon \le r_2$. However, any $x$ that satisfies $x - \epsilon \le r_1$ immediately satisfies $x - \epsilon \le r_2$, as $r_1 < r_2$, meaning that the same perturbation induces an error on both classifiers. In conclusion, if condition 1) is satisfied, 2) cannot be.
> > >
> > > For the case of general parallel sets $A^{r_1}, A^{r_2}$, note that if $r_1 < r_2$ then $A^{r_1} \subset A^{r_2}$. Then the argument is similar to the example above.

---

> > > > ### Comment · Reviewer_Tq42 · 2023-08-15
> > > >
> > > > Interesting!
> > > > That's a good resolution to the measurability issue.
> > > > I will keep my score and vote for acceptance.

---

### Official Review · Reviewer_gWM1 · 2023-07-03

**Soundness:** 3 good
**Presentation:** 3 good
**Contribution:** 3 good
**Rating:** 7
**Confidence:** 2

**Summary:**

This paper provides an analysis of the role of randomization in developing adversarially robust classifiers. It explores the conditions under which randomized ensembles outperform deterministic classifiers regarding adversarial risk. The study also highlights the existence of deterministic classifiers that is at least the same adversarial risk as the probabilistic model, where the authors also connect the theory to weighted ensembles and randomized smoothing. Overall, the paper offers valuable insights into randomization for building robust classifiers in adversarial settings.

**Strengths:**

1. The paper is well-written, ensuring clarity and ease of understanding for readers.

2. The paper offers a foundational understanding of the significance of randomization in building robust models. Especially, I find constructing a probabilistic model from a deterministic model and back particularly interesting.

3. The paper includes an insightful analysis of existing methods (e.g., randomized smoothing ), showcasing their effectiveness.

**Weaknesses:**

1. While the theoretical analysis presented in this paper is intriguing, it leaves me questioning its practical implications in constructing robust classifiers. Specifically, I am concerned about how these theoretical findings translate into practical strategies for building classifiers that perform effectively in real-world scenarios. For example, Theorem 4, in its current form, only addresses the binary case, limiting its applicability. I think the authors can add more discussion on how to guide the future empirical robust model design.

2. The authors show there exists "a probabilistic classifier that is at least as robust as the best deterministic classifier." However, I think it is more important to guide future research to find such a better probabilistic classifier in a principled way.

3. Minor: It is recommended to explain the term "Borel σ-algebra."
4. Minor: Moving the fig1 to near page 4 is better.

As I am not an expert in theoretical research, evaluating the novelty and contribution is hard for me.


**Questions:**

1. In adversarial literature, we might consider the attackers can submit malicious input multiple times, where one misclassification is considered a "success attack." I believe the probabilistic model might be more likely to be broken. What happens to the current theorem if considering such an attacking scenario? I found a recent work [1] which is considering this issue.

2. This paper considers a BHS composed of infinitely many linear classifiers. How does the theorem change if BHS is composed of non-linear classifiers?

3. The paper mainly discussed robustness. How does the clean performance change of building a probabilistic model from deterministic models and back?

[1] Lucas et al. Randomness in ML Defenses Helps Persistent Attackers and Hinders Evaluators. https://arxiv.org/pdf/2302.13464.pdf

**Limitations:**

The authors are suggested to include a paragraph discussing the limitations of the paper.

---

> ### Author Rebuttal · Authors · 2023-08-09
>
> We would like to thank you for your review. We are very happy to know that you find that our work offers a foundational understanding of the significance of randomization in building robust models. Furthermore, we also agree with you in that this theoretical contribution should be followed by practical ones. Particularly, we are very eager to work on practical algorithms to optimize the *matching penny gap* of a set of classifiers. Below we address your remarks and questions.
>
> > In adversarial literature, we might consider the attackers can submit malicious input multiple times, where one misclassification is considered a "success attack."
>
> Thank you for the interesting reference. We found out that it shares some ideas with our work. More precisely, their conclusion seems to agree with our conclusion in Section 4, where taking out the noise of a probabilistic classifier could give a potentially better deterministic one. However, they conclude this under a threat model that is particularly pessimistic for randomized models.
>
> With the definitions we use for probabilistic classifiers, at any point $(x, y)$ we have that $\mathbf{h}(x)_y$, the $y-$th component of $\mathbf{h}(x)$, is the probability that we predict the correct label for $x$. In general, these probabilities will lie between 0 and 1, without being exactly 0 or 1, which means that simply making $n$ inferences on the same point, without even computing an attack, will decrease the chance of correctly predicting $n$ times in a row. Under this threat model, the performance of a probabilistic classifier will degrade as $n$ gets bigger.
>
> Consider a point $(x, y)$ and $M$ classifiers that are in a *matching penny* configuration, that is, **they are all individually vulnerable, but no attack can fool two classifiers simultaneously**. Recall that this is the scenario in which a mixture of classifiers provides the greatest gains in robustness w.r.t the original $M$ models considered. All of the $M$ original models have 0 accuracy on $(x,y)$ (under attack). However, using the mixture of the $M$ models gives an expected accuracy of $\frac{M-1}{M}$ because no matter which classifiers the attacker chooses, there is a probability of $\frac{M-1}{M}$ that we use another model to predict. This represents a gain of $\frac{M-1}{M}$ in expected accuracy. Now, under the threat model with $n$ inference passes, this gain passes from $\frac{M-1}{M}$ to $(\frac{M-1}{M})^n$, which will decrease rapidly as $n$ increases. As an example, if $M=5$ and $n=5$, the gain passes from $0.8$ to $0.32$. If $n=10$, then it is only $0.107$.
>
> On the other side of the spectrum, adding $n$ is also detrimental for those cases in which considering a mixture actually *added a vulnerability*. Just as the gains of the mixture become less important, also the new vulnerabilities become more critical. To see this, imagine the situation in which at a point $(x, y)$ there are $M-1$ models that are robust, and only 1 classifier that is vulnerable. An optimal attacker will attack this single model, as it is the only vulnerable one. In our setting, the best model had an accuracy of 1 at the point, and the mixture had an expected accuracy of $\frac{M-1}{M}$, meaning the mixture introduced a loss in performance of $\frac{1}{M}$ at the point. Now if we introduce $n$ and the attacker proposes the same attack, the mixture will have an expected accuracy of $(\frac{M-1}{M})^n$, meaning the vulnerability introduced by the mixture is now $1 - (\frac{M-1}{M})^n$, which tends to 1 as $n$ gets bigger.
>
> In conclusion, our results show that mixing can be beneficial in certain cases, but if one considers this new, more pessimistic threat model parametrized by $n$, the gains provided by mixtures decrease, and the vulnerabilities introduced increase, making it harder for the mixture to bring any improvement w.r.t the original deterministic classifiers.
>
> > How does the clean performance change of building a probabilistic model from deterministic models and back?
>
> The clean risk of a mixture equals the average risk of the base classifiers. There is no gain nor loss of accuracy when considering mixtures. This can be seen in the following equation:
>
> $$\\mathcal{R}(\\mathbf{h}_{\\mu}) = \\mathbb{E}\_{x, y \\sim \\rho} \\left[ \\mathbb{E}\_{h \\sim \\mu} \\left[ \\mathcal{1}\\{h(x) \\ne y \\} \\right] \\right] =  \\mathbb{E}\_{h \\sim \\mu} \\left[ \\mathbb{E}\_{x, y \\sim \\rho} \\left[ \\mathcal{1}\\{h(x) \\ne y \\} \\right] \\right] =  \\mathbb{E}\_{h \\sim \\mu} \\left[ \\mathcal{R}(h) \\right] $$
>
> The average risk is lower bounded by the lowest risk achievable in $\mathcal{H}_b$. This means that mixing does not offer any gain in standard risk, which is in contrast with what is shown in Section 3, where we study the conditions under which mixing does improve adversarial risk.
>
> When it comes to Theorem 4.1, the proof can be replicated with equality, meaning that for a binary probabilistic classifier $\mathbf{h}: \mathcal{X} \to \left[0, 1 \right]$, it holds that $\mathcal{R}(\mathbf{h}) = \int_0^1 \mathcal{R}(h^{\alpha}) d\alpha \ge \inf_{\alpha} \mathcal{R}(h^{\alpha})$. So we can say that inside the same family of $\alpha$-threshold classifiers, there exists one with better standard risk than the original probabilistic classifier.
>
> > How does the theorem change if BHS is composed of non-linear classifiers?
>
> We do not consider linear classifiers only, apart from certain examples, like Example 1 or Figure 1. The results that we prove hold for non-linear classifiers too. We agree we need to add some explicit assumption on the family of classifiers $\mathcal{H}_b$, but by no means is this family restricted to linear classifiers. Our work covers many practical scenarios of interest, like neural networks with continuous activations and many other parametric families of hypothesis.
>
> [1] Lucas et al. Randomness in ML Defenses Helps Persistent Attackers and Hinders Evaluators.

---

> > ### Comment · Reviewer_gWM1 · 2023-08-14
> >
> > Thank the authors for the rebuttal. I appreciate the detailed explanation. I hope the authors can incorporate discussion in the rebuttal to improve the paper. It is encouraged to mention [1] and discuss the differences and similarities. I think this can make the paper to be accessed by more audiences. I increase my score to 7.
> >
> > [1] Lucas et al. Randomness in ML Defenses Helps Persistent Attackers and Hinders Evaluators.

---

> > > ### Author Response · Authors · 2023-08-14
> > > **Thank you**
> > >
> > > We are very honoured by your appreciation or our paper.
> > >
> > > We forgot to include it explicitly in our first answer, but indeed we will be including the discussion about the threat model considered in our work. Here we will discuss [1], as it shows that the role of randomization can change depending on the chosen threat model and the hypothesis about the attacker's capabilities.

---

### Official Review · Reviewer_TdsR · 2023-07-27

**Soundness:** 3 good
**Presentation:** 3 good
**Contribution:** 3 good
**Rating:** 6
**Confidence:** 3

**Summary:**

The paper addresses the theoretical problem of how randomization affects adversarial robustness. It establishes a condition that is both necessary and sufficient for enhancing the robustness of probabilistic models. Furthermore, it proves that in binary classification, a deterministic model can always achieve at least the same level of robustness. The paper also discusses the crucial condition for the effectiveness of randomization. Although the assumptions are strict, the theories in this work can serve as a foundation to further investigate adversarial robustness in probabilistic classifiers.

**Strengths:**

1. The paper is well-written and well-organized.
2. The paper works on an interesting theoretical problem of the effect of using randomization on adversarial robustness. The theoretical result gives a promising hint to better understand the randomization methods.

**Weaknesses:**

1. The theories and proofs in this work heavily rely on binary classifiers with a misleading title. This restricts the practical application of these theories in real randomization scenarios.
2. The work presented lacks in applying and evaluating the theory, even in binary cases. It does not explore the potential of the proposed theory to improve randomization or identify scenarios where a deterministic classifier is possible. For instance, can we find or estimate better deterministic models corresponding to baseline probabilistic classifiers? How can we apply the theories to enhance adversarial robustness? Including quantitative evaluations would significantly improve the quality of the work.


**Questions:**

N.A.

**Limitations:**

See weakness

---

> ### Author Rebuttal · Authors · 2023-08-09
>
> Thank you for your review. We agree with you in that a very interesting path for future work is the practical implementation of this theory. We are very interested in exploring how we can actually boost adversarial robustness, either by considering the right probabilistic classifiers (that maximize the *matching penny gap*), or by better understanding the threshold step used in Section 4 for binary classifiers. We believe that this theory should definitely give intuitions on practical actions towards implementable robust classifiers.
>
> Below, we adress some of your remarks in more detail.
>
>
> 1. Section 3 does not assume binary classification, it also works in multiclass. It is Section 4 which is restricted to binary classification.
>
> 2. We provide theoretical evidence that we can indeed estimate the better deterministic models by training weighted ensembles and randomized smoothing classifiers in opposition to randomized ensembles and any kind of noise injection classifiers, which are abundant in the literature.
>
> 3. The question of enhancing adversarial robustness is very interesting, and the theory we have developed can motivate new methods that rely on diversity promotion. It should also motivate the crafting of more powerful attacks that can adapt to the mixture scenario, i.e. attacks that maximize the mass of simultaneously attacked classifiers.

---

> > ### Comment · Reviewer_TdsR · 2023-08-16
> >
> > Thank you for your response. It has addressed all of my concerns, and the work has the potential for further investigation. I will increase my score to 6.

---

### Author Rebuttal · Authors · 2023-08-09

We thank all the reviewers for their very valuable comments and questions. Many of them also pointed out interesting references that sparked interesting discussions within the team. We are glad that multiple reviewers found that our work tackles an interesting question related to adversarial robustness, and that it should contribute to future work on the use of randomized classifiers for adversarial robustness. Below we address comments that were raised by multiple reviewers

> About the term *matching penny gap*

We agree that it would be better to add a more complete explanation on what the term means and why it is relevant to our analysis. Below is a more in depth explanation that we would like to include in the Supplementary material, and a summary of it if possible in the main paper, Section 3 with the introduction of the matching penny gap.

Recall that in the original matching penny game between player 1 (attacker) and player 2 (defense), each player has a penny coin and has to secretly position its penny in either heads or tails. Then, both coins are simultaneously revealed and the outcome of the game is decided as follows: attacker wins if both pennies match. If they do not match, then defender wins.

The parallel with our work can be simply explained as follows: Take a point $(x, y)$ and two classifiers $tails, heads$ that correctly classify $x$. Suppose that both $tails$ and $heads$ are vulnerable at $x$ for some given threat model, but add the key assumption that they cannot be attacked at the same time (Figure 1.a). That is, even though an optimal attacker can fool each classifier individually, there is no point in the allowed perturbation region $B\_{\epsilon}(x)$ in which both are simultaneously fooled. Consider that the defender is using a randomized ensemble that picks either $tails$ or $heads$ at random for inference. This was introduced in [1] with the name *mixtures of classifiers*, related to mixed strategies in the context of game theory. In such setting, the optimal attacker that faces such mixture is now in a matching pennies game situation. At each inference pass, the attacker must choose which classifier to attack in order to craft the adversarial example $x'$, and if the chosen classifier matches with the one the defender used, then the attacker wins. If they do not match, the prediction made by the defender will be correct and the attacker would have lost.

Notice that if we increase the number of choices (classifiers) in matching pennies configuration to $M>2$, the game becomes harder for the attacker. Indeed, for every possible choice the defender can make, there is one out of $M$ choices that lead to a successful attack, and $M-1$ that lead to a correct classification. An extreme example of such benefit to the defender is shown in Example 1.

> Completeness of the supplementary material

We agree that adding deeper explanations of examples and the calculations of Figure 1 would increase the overall quality and completeness of the paper. We are going to add these detailed explanations and calculations, as well as the modified proof of Theorem 3.2 with a maximizing sequence. We will also add the comment related to the measurability of the 0-1 loss under attack. Other things we will add :

- Motivation of the name *matching penny gap*
- Adversarial risk for deterministic classifiers (line 102) as ZKUe suggests.

> Figure placement

We will do our best to ensure Figure 1 appears closer to the relevant section.

***

**Supplementary pdf**

In the supplementary PDF page we added three plots that show how the adversarial risk of the deterministic classifiers $h^{\alpha}$ behaves w.r.t $\alpha$, and compare it with the adversarial risk of the base probabilistic classifier $\mathbf{h}$. These plots show that, following Theorem 4.1, there is always some $\alpha$ such that $h^{\alpha}$ performs better that $\mathbf{h}$. The dataset used was CIFAR-10, where classes were grouped into `animal` and `vehicle` to create a binary classification task, creating a 60%-40% class distribution.


[1] Randomization matters. How to defend against strong adversarial attacks

---

### Decision · Program_Chairs · 2023-09-21

**Decision:**

Accept (spotlight)

**Comment:**

## Overview

This is a theory paper studying randomness and adversarial robustness. One of the main results is a condition for improving robustness in a randomized defender setting. As a consequence, the authors show that deterministic models can be at least as robust as randomized models for binary classification. The authors study multiple approaches to constructing robust randomized classifiers, e.g., randomized ensemble classifier (REC), weight-noise injection classifier (WNC), input-noise injection classifier (INC). The authors analyze these approaches in the context of their performance relative to the best deterministic classifier, deriving a number of new results.

On the positive side, the reviewers found the results to be interesting, novel, and insightful. The theoretical results are also motivated by common approaches in practice, leading to potential future empirical impact. Given that the authors provide new results on basic, fundamental questions, there is likely be a broad scope for future work inspired by this paper.

On the negative side, there are some serious concerns around the exposition and clarity of the writing (see below in the improvements section).

The authors and reviewers had a very fruitful discussions. In particular, several very technical questions and suggestions were raised by the reviewers. The authors have taken these into consideration, and I believe they will improve the final version of the paper. The discussion and clarifications also led to a number of score increases, speaking to the impact and correctness of the paper.

## Improvements:

It would be good to continue improving the final version of the paper by including (either in the main paper or appendix) more exposition around the main results in the paper.  For example, in the recent comments, there a number of improvements that should be included in the final version of the paper (e.g., the name matching penny gap and adversarial risk for deterministic classifiers). Please also add examples/calculations relating to Figure 1 and any clarifications around the theorem statements and proofs. Similarly, it is quite important to be clear about the attack model: access to the parameters, gradients, but not the defender randomness.